# Netrin-1 disrupt high-fat-diet-induced adipogenesis via the PPARγ and Wnt/β-catenin signaling pathways
Hang Shi[1,2], Jianghai Tang[2], Xuemiao Yan[3], Tingyu Ke[4], Duo Mao [1] ✉, Deling Kong [5] & Chen Li [2] ✉

The present study reports a detrimental role of adipose-derived Netrin-1 in adipose remodeling. Following an 8-week high-fat feeding period of male transgenic mice lacking adipose Netrin-1 expression (*Ntn1*[AKO]), improved metabolic parameters were observed, accompanied by systemic weight gain and increased inguinal white adipose tissue (WAT) mass. The *Ntn1*[AKO] preadipocytes exhibit increased cell proliferation with decreased collagen deposition. WAT Netrin-1 overexpression using adeno-associated virus (with an aP2 promoter) results in impaired glucose tolerance in both high fat and normal chow-fed mice. Netrin-1 overexpression attenuates adipogenesis via inhibition of the PPARγ activity and activation of the Wnt/β-catenin pathway. Moreover, Netrin-1 is directly responsive to the hypoxic regulator HIF-1α in both adipocytes and preadipocytes. The present study suggests that Netrin-1 disrupts adipogenesis and adipocyte function by inhibiting compensatory adipose remodeling during excessive calorie intake and may be considered a potential therapeutic target for high fat diet-induced obesity and type 2 diabetes.

Type 2 diabetes (T2D) accounts for approximately 90% of all diabetic individuals[1]. An imbalance between energy intake and energy expenditure is one of the main causes of obesity and a major risk factor for T2D, often resulting in the accumulation of excessive fat and adipose tissue expansion. Although occurrences of adipose expansion are often pathological, likely resulting in the subsequent onset of metabolic disorders, it has also been shown that increase in adipocyte numbers could, in fact, alleviate the energy imbalance brought about by excess calorie intake as a compensatory approach. Pathological adipose tissue expansion typically manifests as preferential accumulation of the visceral adipose tissue, suppression of adipogenic differentiation, and dysfunctional adipocytes. Whereas metabolically healthy adipose expansion is characterized by preferential accumulation of subcutaneous adipose tissue and enhanced adipogenic differentiation capacity[2]. Accumulation and expansion of the visceral white adipose tissues (WATs) are also highly associated with the development of insulin resistance and is considered an independent risk factor for T2D[3]. On the other hand, accumulation of WATs in the subcutis has been shown to exert protective benefit against the onset of insulin resistance in individuals with comparable body weights[4].

Indeed, the lipogenic differentiation ability of the preadipocytes is essential for the maintenance of adipose function and systemic metabolic balance[5]. As previously reported in animal studies, high-fat-fed mice with impaired lipogenic differentiation capability had severely compromised glucose tolerance and insulin responsiveness, albeit exhibiting lean phenotype[6]. Limited subcutaneous adipogenesis was also observed in non-obese T2D individuals, supporting a direct association between imbalanced adipose function and metabolic disorders in human. Earlier publications based on observations made using genetically modified transgenic animals have already demonstrated improvement of metabolic health by enhancing preadipocyte differentiation and adipose expansion[7]. Regulatory proteins that are related to adipogenesis, e.g., the heparan sulfate 6-O-sulfo-transferase, have also been reported to promote a healthy obese metabolic phenotype[8]. However, the causal relationship between adipose lipid differentiation and metabolically healthy fat remodeling remains unclear.

Netrin-1 is a secreted protein traditionally known to guide the growth and localization of neurons during developmental processes[9]. Indeed, Netrin-1 has been shown to participate in tumor initiation and progression by arresting malignant cell apoptosis. Tumor neoangiogenesis[10], vascular

[1]Precision Medicine Institute, The First Affiliated Hospital of Sun Yat-Sen University, Sun Yat-Sen University, Guangzhou, China. [2]Institute of Biomedical Engineering, Chinese Academy of Medical Sciences & Peking Union Medical College, Tianjin, China. [3]Department of Pathology, Tianjin First Center Hospital, Tianjin, China. [4]Department of Endocrinology, The Second Affiliated Hospital of Kunming Medical University, Kunming, Yunnan, China. [5]State Key Laboratory of Medical Chemical Biology, Key Laboratory of Bioactive Materials, Ministry of Education, and College of Life Sciences, Nankai University, Tianjin, China. ✉e-mail: maod6@mail.sysu.edu.cn; cli@bme.pumc.edu.cn

diseases[11] and metabolism[12] are also facilitated by Netrin-1 as reported. Furthermore, it has also been reported that blockade of Netrin-1 could inhibit the epithelial-to-mesenchymal transition of tumor cells, resulting in lessened resistance to anti-cancer treatment, such as chemotherapy[13]. Meanwhile, it was shown that the macrophage-derived Netrin-1 is responsible for the progression of lung fibrosis[11] and formation of aortic aneurysm[14]. Elimination of hematopoietic Netrin-1 has also been reported to attenuate macrophage retention in the atherosclerosis plaques[15] as well as in the adipose tissues extracted from obese mice, leading to improved systemic insulin sensitivity[12]. Considering the seemingly different roles of Netrin-1 in different tissue types and the implicated effects of Netrin-1 on adipose metabolism in particular, this study aimed to identify the potential role and underline mechanisms of Netrin-1 in the adipose tissue and subsequent effects on systemic metabolic regulation.

## Results

### Adipose Netrin-1 deletion restores high-fat-induced systemic metabolic dysfunction via enhanced adipose tissue remodeling

Given the multiple roles of Netrin-1 in physiological processes, especially in metabolism, we first obtained the adipose Netrin-1 knockout mice using an aP2 Cre-lox system $Ntn1^{f/f}$: aP2-Cre (abbreviated as $Ntn1^{AKO}$) (Fig. S1A). Immunoblotting analysis verified the downregulation of Netrin-1 expression in the brown adipose tissues (BATs) and WATs (Fig. S1B), but not in other tissue types, such as the kidney and brain (Fig. S1C).

The $Ntn1^{AKO}$ mice were then maintained on a HFD or chow diet for 8 weeks, during which the metabolic parameters, including body weights, fasting plasma glucose levels, glucose tolerance and insulin responsiveness were measured and recorded. As shown in Fig. 1A, B, for the experimental groups that were maintained on normal chow diet, the average body weights and fasting plasma glucose levels of both Flox (Flox, shown in black) and $Ntn1^{AKO}$ mice (ako, shown in red) were comparable throughout the observation period. Glucose tolerance (Fig. 1C) and insulin responsiveness (Fig. 1D) were also similar between the chow-fed $Ntn1^{AKO}$ and Flox groups, indicating no significant effect of Netrin-1 under chow-fed condition.

Importantly, however, for the high-fat-fed groups, mice with adipose deficiency of Netrin-1 ($Ntn1^{AKO}$) exhibited significantly better glucose tolerance (Fig. 1E) and insulin sensitivity (Fig. 1F). Reduced fasting plasma glucose levels were also recorded from the high-fat-fed $Ntn1^{AKO}$ group (Fig. 1G). The improved glucose tolerance and insulin responsiveness in the high-fat-fed $Ntn1^{AKO}$ mice could, at least partly, be explained by enhanced insulin signaling activity, as demonstrated by increased phosphorylation of Ser473-AKT in insulin-exposed mature primary adipocytes extracted from the $Ntn1^{AKO}$ mice (Fig. S2A, B). The improved glucose metabolism was also accompanied by lower serum triglyceride (Fig. 1H) and free fatty acid content (Fig. 1I), implicating the potential impact of Netrin-1 on glucose and lipid metabolism under high-fat condition.

Besides, in the present study, we also observed an unexpected albeit significant increase of body weight from the high-fat-fed $Ntn1^{AKO}$ mice when compared to the Flox group of the same high-fat feeding condition (Fig. 1J, K). This could perhaps be regarded as a compensatory effect under high-fat-feeding conditions.

Indeed, previous publications have shown that healthy adipose tissue remodeling (such as adipogenesis) in response to metabolic dysregulation could paradoxically offer protection in maintaining metabolic balance by storing excessive fat while exerting anti-diabetic benefit[5]. Moreover, impeded adipogenesis and lipid re-distribution have also been reported to associate with the development of insulin resistance and T2D[16–18]. Given the previously published findings, magnetic resonance imaging (MRI) body composition scanning was subsequently employed to examine whole-body mass distribution of the high-fat-fed $Ntn1^{AKO}$ and Flox mice. It is evident that the body weight gain observed from the high-fat-fed $Ntn1^{AKO}$ mice is mainly attributable to fat mass increase (Fig. 1L). Specifically, a distinct increase of the iWATs mass was seen in the high-fat-fed $Ntn1^{AKO}$ mice (Fig. 1M, N).

No differences in other segments of the adipose tissue were detected, suggesting that adipose Netrin-1 deficiency promoted iWAT expansion only under high-fat feeding conditions. Regarding the chow-fed mice, similar to the results of metabolic parameters presented in Fig. 1A–D, no differences in fat or lean mass accumulation nor body weight were observed between the $Ntn1^{AKO}$ and Flox mice (Fig. S3A, B).

Subsequent H&E staining and comprehensive metabolomics profiling of the WATs from the $Ntn1^{AKO}$ and Flox mice revealed no significant difference in adipocyte size or metabolites composition between the high-fat-fed $Ntn1^{AKO}$ and Flox mice (Fig. S3C–E), implicating that the iWAT expansion as a result of adipose Netrin-1 deficiency was not due to adipocyte enlargement. A significant reduction in Edu$^+$ incorporation was observed in Netrin-1 overexpressing 3T3-L1 cells (Fig. 1O, P). We also observed elevated Netrin-1 expression in the SVFs that were isolated from the iWATs as compared to the eWATs (Fig. S3F) of normal chow-fed Flox mice. These results indicate that Netrin-1 overexpression likely impeded preadipocyte proliferation, whereas deficiency of adipose Netrin-1 enhanced iWAT expansion by promoting preadipocyte proliferation in young $Ntn1^{AKO}$ mice.

Limited WAT fibrosis is another feature of WAT remodeling and healthy adipose function[19,20]. By picrosirius red staining and qPCR, we found that adipose Netrin-1 deficiency improved adipose collagen deposition with reduced expression of fibrosis-related genes in the WATs (Fig. S4A, B).

In addition, we also observed ameliorated ectopic lipid deposition in the liver samples obtained from the high-fat-fed $Ntn1^{AKO}$ mice (Fig. 1Q), albeit no detectable difference in liver mass was detected between the $Ntn1^{AKO}$ and Flox groups. Taken together, our results so far demonstrate that adipose Netrin-1 deficiency led to healthy WAT remodeling and improved metabolic state during high-fat feeding.

### Upregulation of adipose Netrin-1 impairs systemic glucose homeostasis in high-fat fed mice

Adipose-targeted overexpression of Netrin-1 was then performed using the adeno-associated virus (AAV) to further clarify the role of adipose Netrin-1. Immunoblotting and qPCR analyses confirmed that the expression of Netrin-1 was stably increased in mouse WATs for 8 weeks following AAV-Netrin-1 delivery (Fig. S5A, B). Considering the potential hepatic accumulation of the AAV vectors as a result of the tail-vein delivery approach, hepatic protein levels of Netrin-1 were also examined, and as shown in the Fig. S5C, no significant Netrin-1 expression could be detected, which could be due to low level of the aP2 expression in the liver[21] as well as a possible low recombination efficiency of Netrin-1.

Similar to our earlier observations using the adipose Netrin-1 deficient $Ntn1^{AKO}$ mice, no significant change was detected regarding the average body weight between the chow-fed adipose Netrin-1-overexpressing AAV-Netrin-1 mice and the controls (Fig. 2A). However, regarding systemic glucose tolerance, the chow-fed AAV-Netrin-1 mice exhibited significantly worsening glucose tolerance despite having similar fasting plasma glucose levels as compared to the controls (Fig. 2B, C).

Effects of high-fat feeding in addition to adipose Netrin-1 overexpression on systemic metabolic parameters were also investigated by maintaining the AAV-Netrin-1 and control mice on a HFD for 3 months. The AAV-Netrin-1 mice showed exacerbated glucose tolerance accompanied by elevated fasting blood glucose levels (Fig. 2E), despite having a moderately lowered average body weight (Fig. 2D). Serum FFA and triglyceride (TG) levels were also elevated in the Netrin-1-overexpressing AAV-Netrin-1 mice (Fig. 2F, G). Additionally, the protein levels of PPARγ were decreased in the iWATs and eWATs of the high-fat-fed AAV-Netrin-1 group (Fig. 2H). Following overexpression of Netrin-1 in adipose tissue, the relative weights of the adipose tissues decreased whereas average weights of the hepatic tissues increased (Fig. S5D) accompanied by increased hepatic lipid content (Fig. S5E), although these changes were lacking statistically significance.

Histological staining results showed no detectable changes regarding the average adipocyte sizes of the iWATs and eWATs following aP2-AAV-targeted adipose Netrin-1 overexpression (Fig. S5F, G), suggesting a lesser

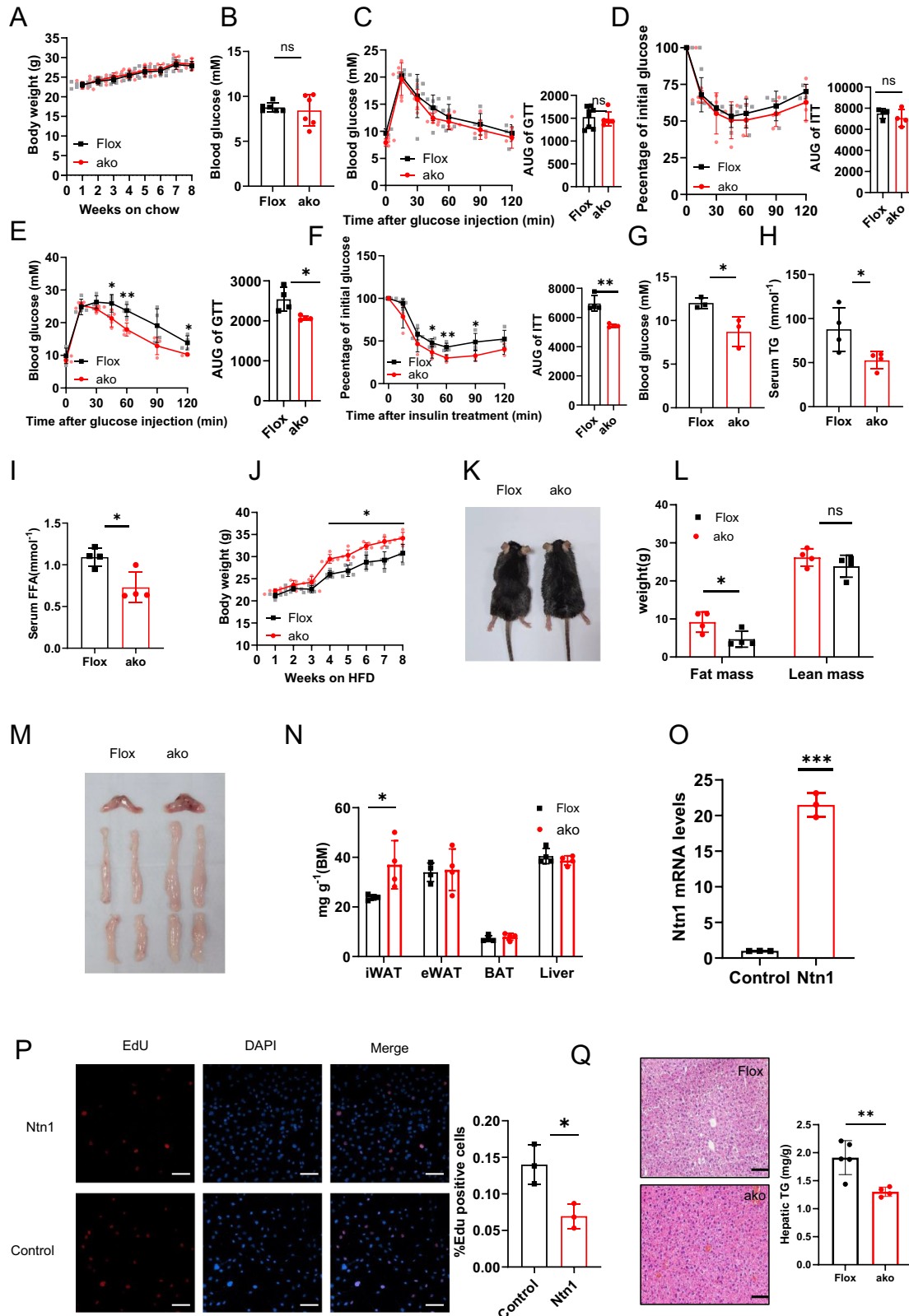

role of adipocyte hyperplasia in Netrin-1-exerted inhibition on adipose expansion.

## Brown adipose tissue (BAT) deficiency does not affect thermogenesis

Since the aP2 promoter affects all types of adipose tissues, the $Ntn1^{AKO}$ mice is also deficient of Netrin-1 expression in the BATs (Fig. S1B). In contrast to the WATs, which is mainly responsible for energy storage, the BATs are known to be involved in metabolism, particularly in energy expenditure via thermogenesis[22]. To examine whether adipose Netrin-1 impacts systemic energy expenditure or thermogenesis, metabolic cages were applied to monitor the metabolic activities of the $Ntn1^{AKO}$ mice. We observed similar average body weights as well as food and water intake between the 8-week-old $Ntn1^{AKO}$ and control animals (Fig. 3A, B). Oxygen consumption and

**Fig. 1 | Adipose Netrin-1 deletion restores high-fat-induced systemic metabolic dysfunction via enhanced adipose tissue remodeling. A** Average body weights of Flox and Ntn1[AKO] mice fed on chow diet for 8 weeks, n = 6 per group. **B** Average 6 h-fasted blood glucose levels of Flox and Ntn1[AKO] mice fed on chow diet for 8 weeks, n = 6 per group. **C** Glucose tolerance tests (GTTs) of Flox and Ntn1[AKO] mice fed on chow diet for 8 weeks, n = 6 per group. **D** Insulin tolerance test (GTT) of Flox and Ntn1[AKO] mice fed on chow diet for 8 weeks, n = 4 per group. **E** GTT of Flox and Ntn1[AKO] mice fed on high-fat diet (HFD) for 8 weeks, n = 4 per group. **F** ITT of Flox and Ntn1[AKO] mice fed on high-fat diet (HFD) for 8 weeks, n = 4 per group. **G** Average 6 h-fasted blood glucose levels of Flox and Ntn1[AKO] mice fed on HFD for 8 weeks, n = 4 per group. **H** Average serum triglyceride level of Flox and Ntn1[AKO] mice fed on HFD for 8 weeks, n = 4 per group. **I** Average serum free fatty acid level of Ntn1[AKO] mice fed on HFD for 8 weeks, n = 4 per group. **J** Average body weights of Flox and Ntn1[AKO] mice fed on HFD for 8 weeks, n = 4 per group. **K** Representative image of Flox and Ntn1[AKO] mice fed on HFD for 8 weeks. **L** Average fat mass and lean mass of Flox and Ntn1[AKO] mice fed on HFD for 8 weeks, n = 4 per group. **M** Representative image of brown adipose tissue (BAT), and inguinal white adipose tissue (iWAT) and epididymal white adipose tissue (eWAT) fat depots from Flox and Ntn1[AKO] mice fed on HFD for 8 weeks. **N** Tissue index (weights of specific tissue normalized to body weight) of iWAT, eWAT, BAT, and liver from Flox and Ntn1[AKO] mice fed on HFD for 8 weeks, n = 4 per group. **O** The mRNA level of Netrin-1 in 3T3L-1 cell after Netrin-1 overexpression. **P** EdU labeling of the Netrin-1-overexpressing 3T3-L1 cells and control cells. **Q** Representative H&E staining images and triglyceride concentration of liver from Flox and Ntn1[AKO] mice fed on HFD for 8 weeks. Data are shown as mean ± s.e.m. *P < 0.05, **P < 0.01 over the control group as analysed by unpaired t-test with Welch's correction. Scale bar = 100 μm.

carbon dioxide production were also similar between the Ntn1[AKO] and control groups (Fig. 3C–F), with no detectable difference in the respiratory exchange ratio (RER; Fig. 3G, H). We also investigated potential changes of the uncoupling protein 1 (UCP1), one of the best-known thermogenic effectors and major mediator for thermogenesis following adipose Netrin-1 deficiency in the BATs[23]. Immunoblotting results using BATs obtained from the Ntn1[AKO] and Flox mice showed no change in the protein expression of UCP1 as illustrated in Fig. 3I. Histological staining analysis also suggested no significant morphological differences in the BAT between the Ntn1[AKO] and Flox mice (Fig. 3J), all of which indicated that adipose Netrin-1 deletion do not play a significant part in affecting the UCP-1-dependent thermogenesis in the BATs.

## Human adipose tissue Netrin-1 expression correlates with obesity and T2D

We have downloaded two GEO databases (GSE27951 and GSE179455), both of which contain data obtained from the WAT biopsies of obese and T2D subjects. After data analysis, a total of 499 and 3519 genes were selected based on P-value and Fold change (Fig. 4A) from the two databases, respectively. We then identified 7 genes that were upregulated both in obesity and T2D (Fig. 4B, C), and the level of Netrin-1 was significantly higher than the other genes (Fig. 4C), suggesting a possible role of adipose Netrin-1 in obesity and T2D. ELISA results showed that diabetic individuals had significantly higher serum Netrin-1 level than the nondiabetic controls (Fig. 4D). Similarity we also observed that the protein level of Netrin-1 was highly up-regulated in mice white adipocytes during HFD feeding (Fig. 4E). These observations all support a potential association between Netrin-1 expression with diabetes and metabolism, in line with data obtained from the animal experiments (Figs. 1–3). Moreover, the expression patterns of the Netrin-1 receptors in human adipose tissue were also analyzed, although no significant differences were observed based on data included in the databases (Fig. 4F, G).

## Netrin-1 inhibit adipogenesis via the PPARγ signaling pathway

To further investigate the potential mechanisms via which Netrin-1 affect WAT remodeling, the WATs were extracted from Ntn1[AKO] and Flox mice for RNA-seq analysis. The results shown by the volcano plot demonstrate altered expression profiles of a multitude genes as a result of adipose Netrin-1 deficiency (Fig. 5A). RNA-seq data were then analyzed by Kyoto encyclopedia of genes and genomes (KEGG) pathway analysis and Gene Ontology analysis (Figs. 5B and S6A). We observed expression changes of genes that are involved in multiple metabolic pathways (Fig. S6B), such as upregulation of genes that are related to lipid metabolism and adipogenic differentiation. Moreover, GO analysis also showed altered expression of genes participating in fat cell differentiation (Fig. 5B). As such, the RNA-seq data obtained from high-fat-fed Ntn1[AKO] mice suggest a significant role of adipose Netrin-1 in lipid metabolism as well as WATs adipogenesis, differentiation and metabolism during excessive calorie intake.

Thus, to confirm the role of Netrin-1 on lipid metabolism, we overexpressed Netrin-1 in 3T3-L1-derived adipocytes in vitro (Fig. S6D) and examined the expression profiles of relevant genes that are associated with triglyceride synthesis and hydrolysis (Fig. S6E). The results showed that overexpression of Netrin-1 in the adipocytes impaired the expression of lipogenesis genes: the adipose triglyceride lipase (Atgl) and the hormone-sensitive lipase (Hsl), in line with a proposed association of Netrin-1 deficiency on glycolipid metabolism (Fig. S6A).

Regarding the mechanism investigation of WAT adipogenesis, we then extracted preadipocytes from the WATs of normal healthy mice and the obtained preadipocytes were then maintained in culture and used for adipogenic differentiation induction. Results obtained by qPCR demonstrate that the mRNA level of Netrin-1 decreased as preadipocytes differentiated toward the adipocyte phenotype, accompanied by increased mRNA expression of adipogenic marker genes including Plin, Pparγ, and Cebpα (Fig. 5C). Immunoblotting data also show decreased expression of Netrin-1 and its receptor Unc5b as the preadipocytes underwent adipogenic differentiation (Fig. 5D). In addition, expression of Netrin-1 in 3T3-L1-derived mature adipocytes as well as primary adipocytes isolated from WAT were also examined. As shown in Fig. 5E, F, the protein levels of Netrin-1 expressed by mature adipocytes were significantly reduced as compared to preadipocytes, all of which suggesting a negative correlation of adipose Netrin-1 expression with adipogenesis, i.e., diminished Netrin-1 expression as adipogenesis progresses. Subsequently, to understand whether Netrin-1 directly affects adipogenesis, we stably overexpressed Netrin-1 in primary preadipocytes (Fig. 5G), which were then repeated for adipogenic differentiation induction. Results obtained following Oil Red O staining demonstrate that Netrin-1 overexpression inhibited preadipocytes adipogenic differentiation, shown as significantly less red staining and reduced number of red-stained cells than the control group (Fig. 5H).

Our results from the KEGG analysis (Fig. S6A) also suggest that Netrin-1 may affect the PPAR signaling pathway. PPARγ is a major transcriptional regulator of adipogenesis, preadipocytes with PPARγ signaling deficiency failed to differentiate into mature adipocytes[24]. As a result, we suspect that Netrin-1 may inhibit adipogenic differentiation via the PPARγ pathway. In order to find out whether the PPARγ signaling cascade is responsible for Netrin-1-inhibited adipogenesis, we first examined the expression of PPARγ in Netrin-1-overexpressing preadipocytes during adipogenic differentiation (Fig. 5I). Immunoblotting data showed that the protein levels of PPARγ were significantly reduced in the Netrin-1-overexpressing preadipocytes (Fig. 5I left). Increased PPARγ expression were then observed in iWATs extracted from the Netrin-1-deficient Ntn1[AKO] mice (Fig. 5I right). Thus, our observation showed inhibited PPARγ expression by Netrin-1 in preadipocytes as well as iWATs (Fig. 5I), and that the inhibition of PPARγ was Netrin-1 dependent because it was alleviated by adipose Netrin-1 knockdown (Fig. 5I right). We also treated the Netrin-1-overexpressing preadipocytes with rosiglitazone, a PPARγ agonist and anti-diabetic agent[25]. As evident in Fig. 5J, addition of rosiglitazone rescued the arrest of adipogenesis in Netrin-1-overexpressing preadipocytes, confirming that Netrin-1-elicited inhibition of adipogenesis is mediated by the PPARγ signaling activity.

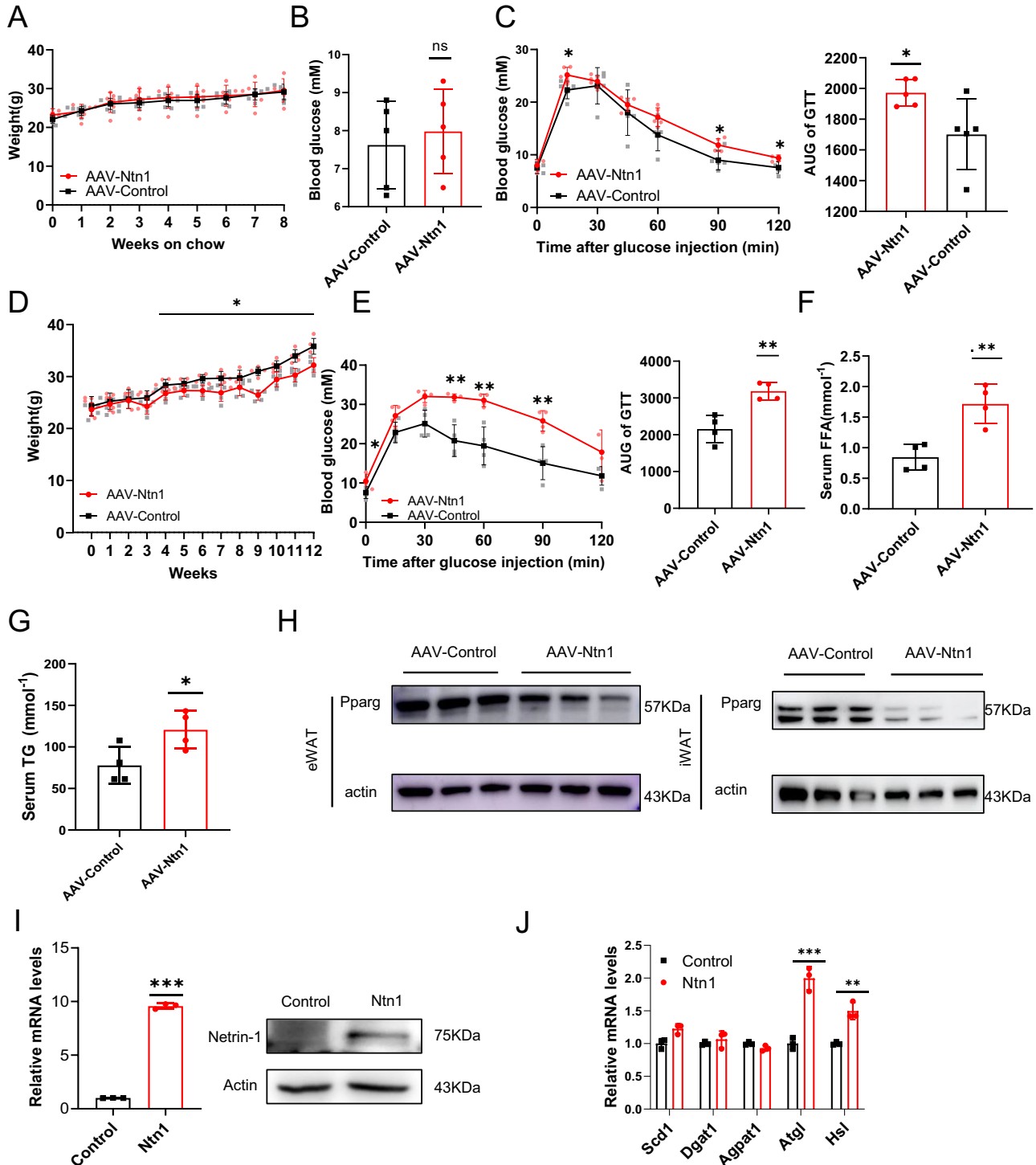

**Fig. 2 | Upregulation of adipose Netrin-1 impaires systemic glucose homeostasis in high-fat fed mice. A** Average body weights of the AAV-Netrin-1-transfected (AAV-Ntn1) and the AAV-control mice that were maintained on normal chow diet for 8 weeks, *n* = 5 per group. **B** Average 6 h-fasted blood glucose levels of the AAV-Ntn1 and AAV-control mice fed on chow diet for 8 weeks, *n* = 5 per group. **C** GTT of AAV-Ntn1 and AAV-control mice fed on chow diet for 8 weeks, *n* = 5 per group. **D** Average body weights of the AAV-Ntn1 and AAV-control mice that were maintained on a HFD diet for 4 months, *n* = 4 per group. **E** GTT of AAV-Ntn1 and AAV-control mice fed on HFD diet after 3 months, *n* = 4 per group. **F** Average

serum triglyceride levels of mice after HFD of 3 months, *n* = 4 per group. **G** Serum free fatty acid level of the AAV-Ntn1 and AAV-control mice after high-fat-fed for 3 months, *n* = 4 per group. **H** The protein level of PPARγin the eWATs and iWATs extracted from the high-fat-fed AAV-Ntn1 and AAV-control mice. **I** The mRNA and protein levels of Netrin-1 following Netrin-1 overexpression in mature adipocytes. **J** The expression profile of lipolysis-related genes following Netrin-1 overexpression in mature adipocytes. Data are shown as mean ± s.e.m. *$P < 0.05$, **$P < 0.01$, ***$P < 0.001$ over control as analysed by unpaired t-test with Welch's correction. Scale bar = 100 μm.

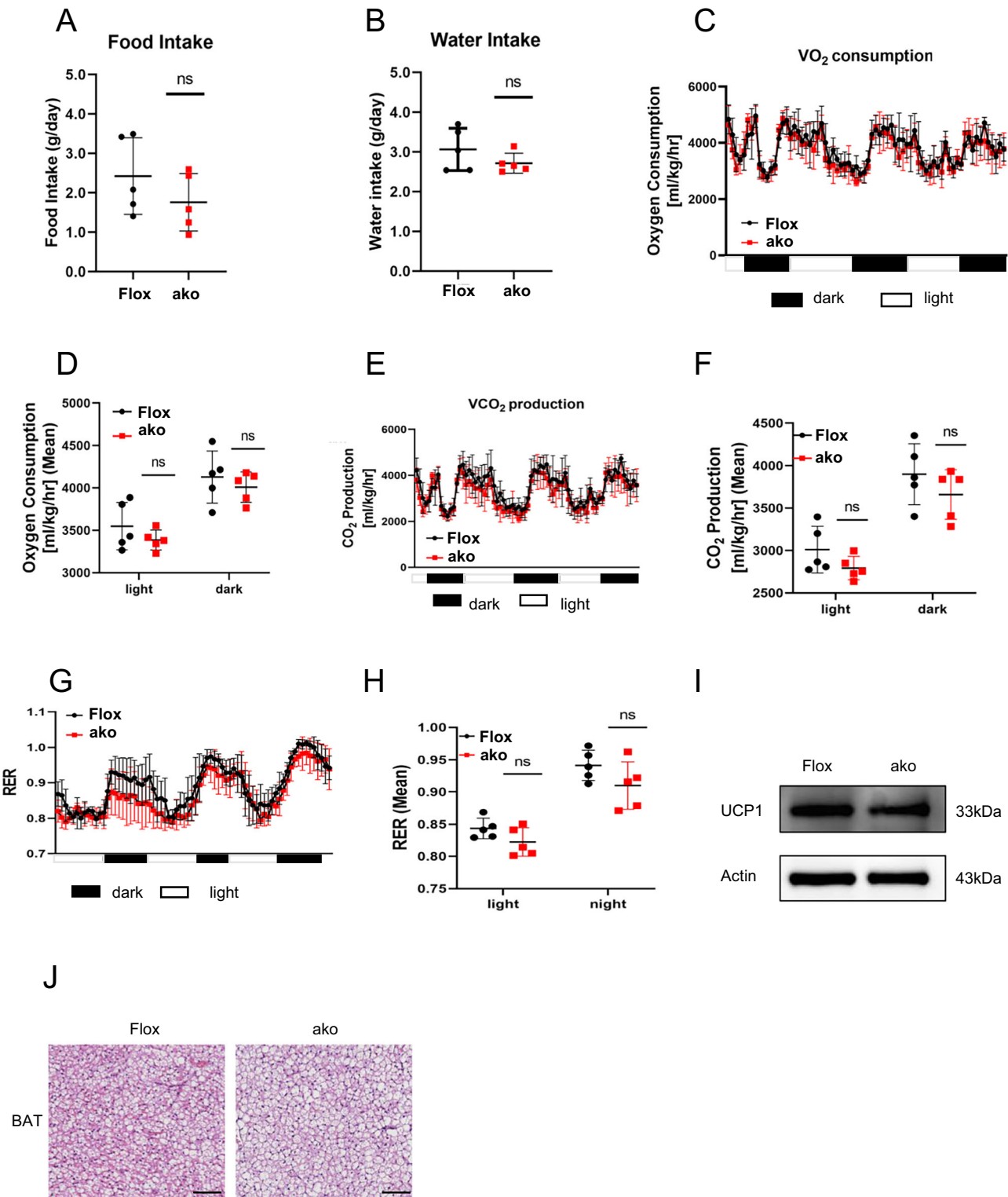

**Fig. 3 | Brown adipose tissue (BAT) deficiency of Netrin-1 does not affect thermogenesis. A** Average food intake of the Flox and $Ntn1^{AKO}$ mice that were maintained on chow diet for 8 weeks, $n = 5$ per group. **B** Average water intake of the Flox and $Ntn1^{AKO}$ mice that were maintained on chow diet for 8 weeks, $n = 5$ per group. Average levels (**C**) and histogram representation (**D**) of $O_2$ consumption of the chow-fed Flox and $Ntn1^{AKO}$ mice during a 3-day light-dark cycle, $n = 5$ per group. Average levels (**E**) and histogram representation (**F**) of $CO_2$ production of the chow-fed Flox and $Ntn1^{AKO}$ mice during a 3-day light-dark cycle, $n = 5$ per group. The average respiratory exchange ratio (**G**) and histogram representation (**H**) of the chow-fed Flox and $Ntn1^{AKO}$ mice, $n = 5$ per group. **I** The protein levels of UCP1 in BATs extracted from the chow-fed Flox and $Ntn1^{AKO}$ mice. **J** Representative images of H&E staining analysis of BATs extracted from the chow-fed Flox and $Ntn1^{AKO}$ mice. Data are shown as mean ± s.e.m. *$P < 0.05$, **$P < 0.01$, ***$P < 0.001$ over control as analysed by upaired t-test with Welch's correction, ns means not significant. Scale bar = 100 μm.

**Fig. 4 | Expression of Netrin-1 in human adipose tissues correlate with obesity and type 2 diabetes.**
**A** Up-regulated genes of GSE27951 and GSE179455 shown in the Venn diagram (log-fold change >2 and *P*-value of <0.05). **B**, **C** Heatmap of overlapping genes from GSE27951 and GSE179455. **D** Serum ELSA level of Netrin-1, diabetic *n* = 25, nondiabetic, *n* = 14. **E** Expression of Netrin-1 in HFD mice mature adipose cells. **F** Expression of Netrin-1 receptors in WATs from nonobese and obese individuals. **G** Expression of Netrin-1 receptors in WATs from obese and type 2 diabetes individuals. Data are shown as mean ± s.e.m. *P < 0.05, ns, nonsignificant over controls as analyzed by upaired t-test with Welch's correction.

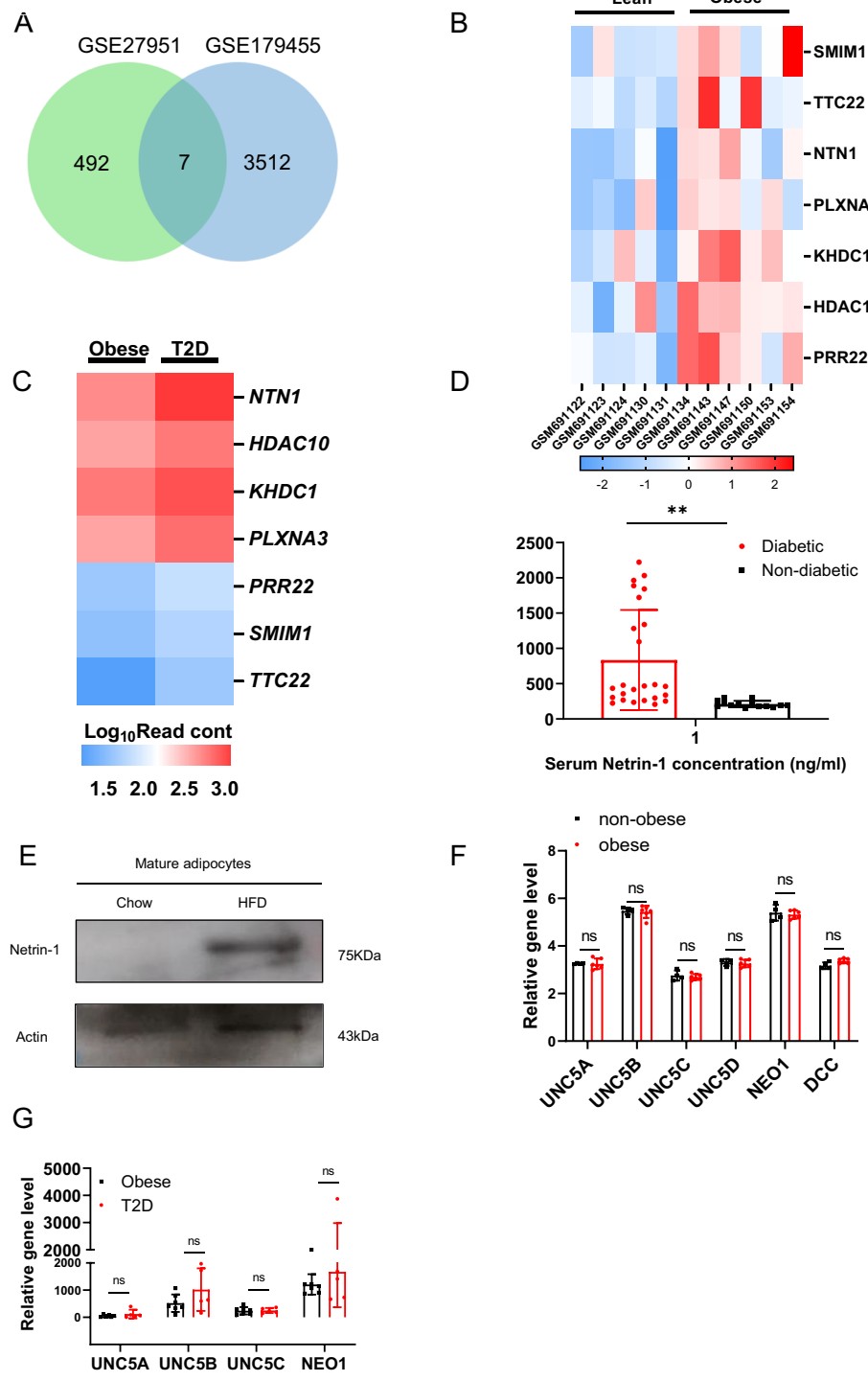

Moreover, decreased hepatic lipid content was observed from the high-fat-fed *Ntn1*[AKO] mice as compared to the Flox controls (Fig. 1Q). Whereas increased, albeit lacking statistical significance, hepatic lipid content was shown from the adipose Netrin-1 overexpressing *Ntn1*-AAV mice (Fig. S5E). Both of the above may also indirectly propose a mechanistic participation of the PPARγ signaling pathway[26,27].

## Netrin-1 activate the Wnt/β-catenin pathway in preadipocytes

Netrin-1 has been reported to activate the Wnt/β-catenin pathway in mouse embryonic stem cell[28]. More relevantly, the PPARγ signaling activity can also be regulated by the Wnt/β-catenin signaling pathway, and inhibition of the Wnt/β-catenin signaling was shown to promote preadipocyte differentiation[29]. To investigate whether Netrin-1 also affects the Wnt/β-catenin pathway, we treated the preadipocyte cell line, 3T3-L1, with recombinant Netrin-1. Immunoblotting data showed that Netrin-1 promoted Gsk3α/β phosphorylation and β-catenin activation in the 3T3-L1 cells (Fig. 5K). Primary preadipocytes were also extracted from the WAT of *Ntn1*[AKO] mice, and we observed significantly mitigated β-catenin activation in these preadipocytes with Netrin-1 deletion (Fig.5L). Moreover, when the mouse primary adipocytes were exposed to Netrin-1 in the presence or absence of a β-catenin inhibitor, XAV939, the inhibitory impact of Netrin-1 on adipogenic differentiation was alleviated by the presence of XAV939 (Fig. S7), confirming the participation of β-catenin downstream of Netrin-1 on the regulation of adipogenesis.

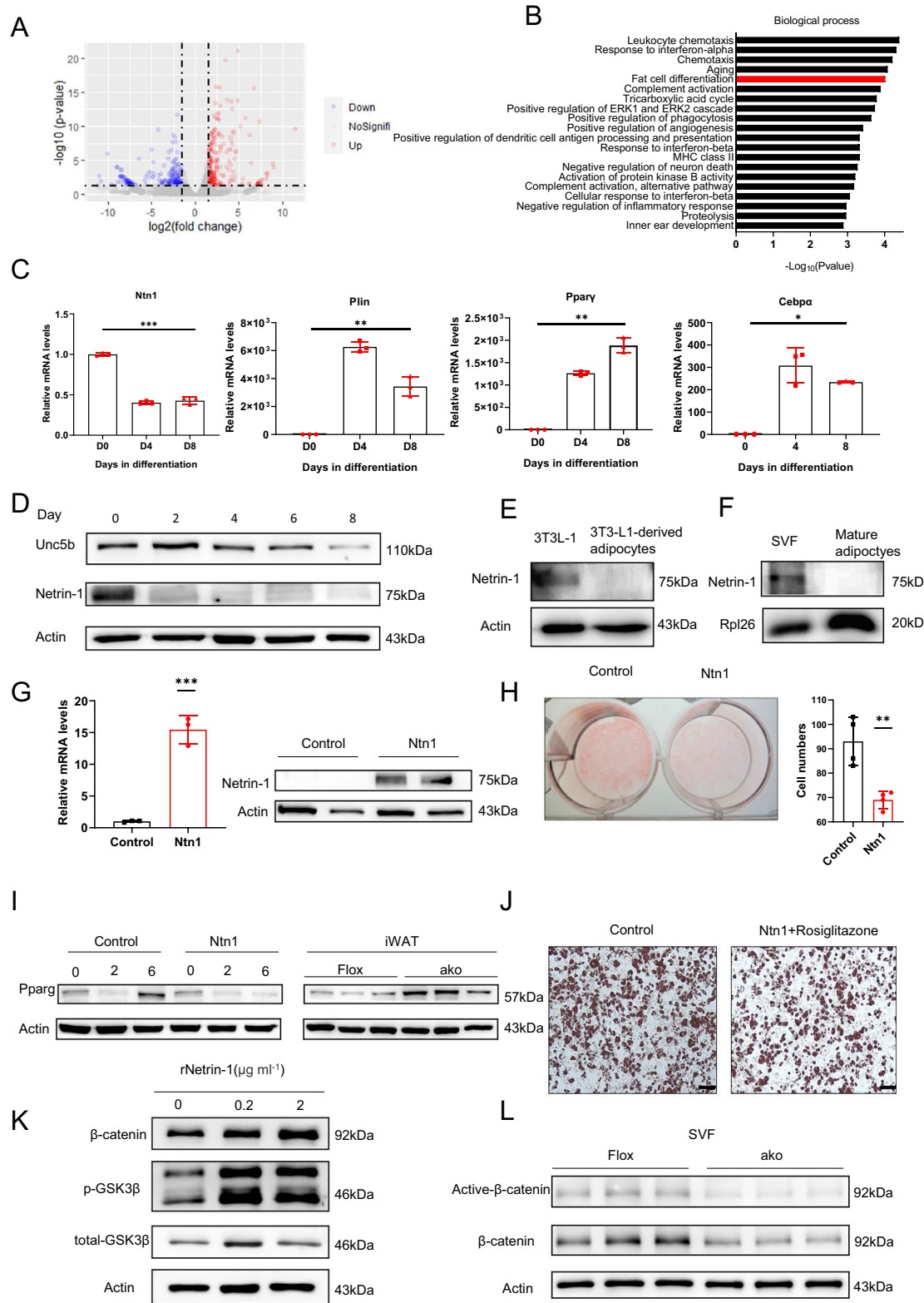

Regarding the mature adipocytes, although previous reports have implicated that the expression and function of β-catenin in mature adipocytes could affect the systemic metabolic phenotype of mice in vivo[30]. However, we did not detect any difference of β-catenin expression in $Ntn1^{AKO}$-origined mature adipocytes lacking Netrin-1 (Fig. S6C).

Here, our results demonstrate that adipose Netrin-1 mitigate adipogenic differentiation of preadipocytes by activating the Wnt/β-catenin pathway. Specifically, it was identified that the Netrin-1-elicited Wnt/β-

catenin activation is achieved by Gsk3α/β inactivation and β-catenin stabilization.

### The hypoxia-inducible factor (HIF-1α) upregulate Netrin-1 expression in adipose tissue and preadipocytes

Pathological adipose expansion often creates a hypoxic environment in the adipose tissue[31]. As reported in animal studies, the presence of local hypoxia could be detected as early as a few days following high-fat feeding[19]. Adipose

**Fig. 5 | Netrin-1 inhibit adipogenesis by inhibition of the PPAR signaling pathway and activation of the Wnt/β-catenin signaling pathway. A** Volcano plot of genes of altered expression (blue means downregulated; red means upregulated) in the WATs of $Ntn1^{AKO}$ mice. **B** GO analysis of WATs from $Ntn1^{AKO}$ mice. Genes involved in fat cell differentiation were enriched and highlighted in red. **C** The mRNA levels of $Ntn1$, $Plin$, $Ppar\gamma$, and $Cebp\alpha$ at day 0, 4, and 8 in primary pre-adipocyte during in vitro adipogenic differentiation induction. **D** The protein levels of Netrin-1 and its receptor Unc5b during primary stromal vascular fractions cells (SVFs) adipogenic differentiation. **E** Protein level of Netrin-1 between 3T3L-1 and 3T3-L1-derived adipocytes. **F** Protein level of SVF and mature adipocytes isolated from WAT. **G** The mRNA and protein levels of Netrin-1 in SVFs (control) and SVFs that overexpress Netrin-1 (Ntn1). **H** Representative images of Oil red O staining and

cell counting results of primary SVFs (control) and SVFs that overexpress Netrin-1 (Ntn1) at day 6 during adipogenic differentiation. **I** The protein level of PPARγ in SVFs that overexpress Netrin-1 (left) and in the iWATs extracted from the Flox and $Ntn1^{AKO}$ mice (right). **J** Representative images of oil red O staining of SVFs (control) and Netrin-1-overexpressing SVFs cultured in the presence of rosiglitazone (Ntn1 + rosiglitazone). Scale bar = 100 μm. **K** The protein levels of β-catenin, phosphorylated-Gsk3β (p-Gsk3β) and total Gsk3β from 3T3-L1 cells that were exposed to increasing dosage of recombinant Netrin-1. **L** The protein levels of active-β-catenin and total β-catenin in primary SVFs extracted from the WAT of Flox and $Ntn1^{AKO}$ (AKO) mice. Data are shown as mean ± s.e.m. *$P < 0.05$, **$P < 0.01$, ***$P < 0.001$. Statistical analysis: for **C**, one-way ANOVA, for **G**, **H** unpaired t-test with Welch's correction.

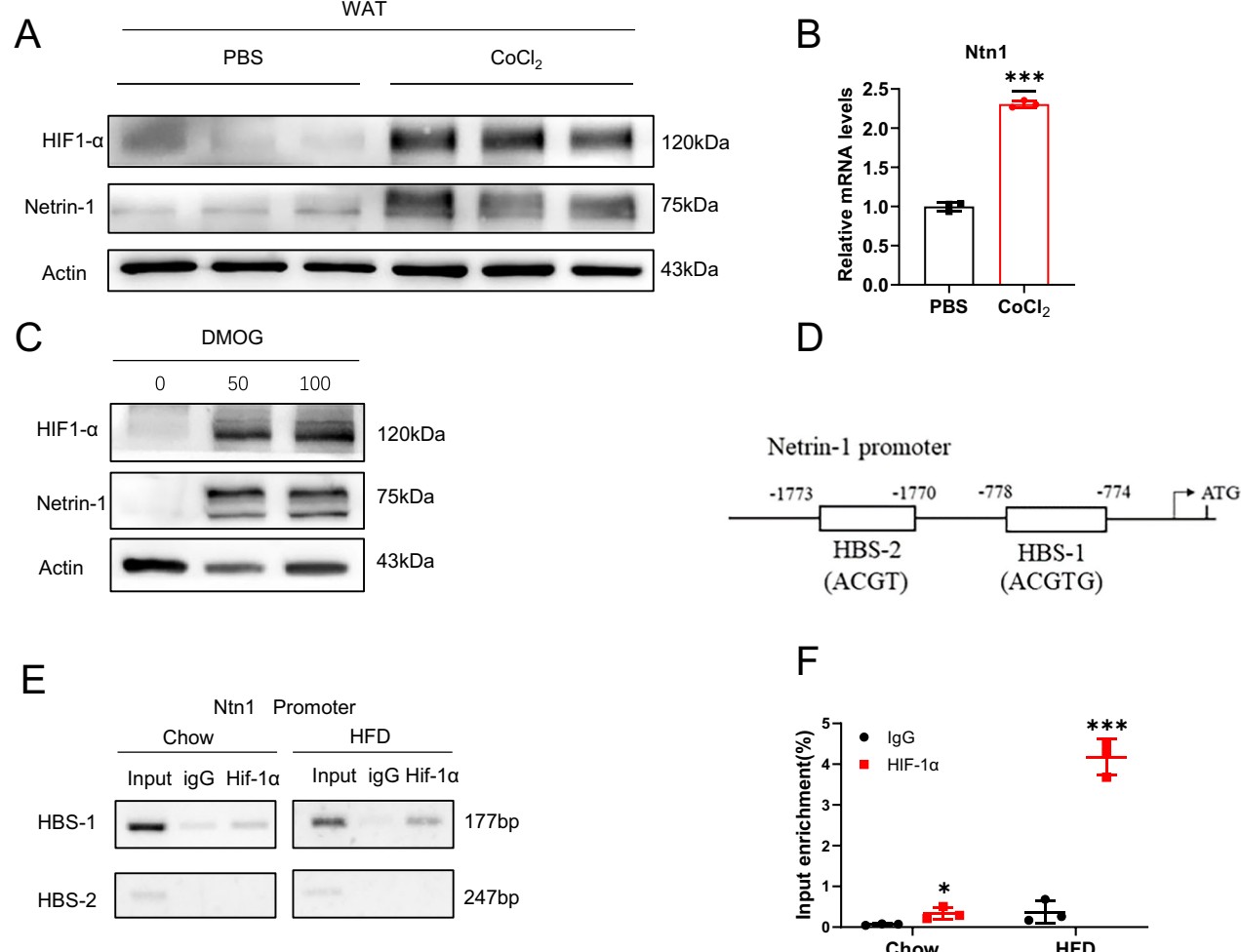

**Fig. 6 | The hypoxia-inducible factor HIF-1α upregulate Netrin-1 expression in adipose tissue and preadipocytes.** The protein (**A**) and mRNA (**B**) levels of HIF-1α and Netrin-1 in primary WATs that were exposed to CoCl₂ or PBS as control. **C** The protein levels of HIF-1α and Netrin-1 in 3T3-L1 cells that were treated with a HIF-1α stablizer DMOG or PBS as control. **D** Schematic representation of the Netrin-1 promoter regions. **E** Results of the ChIP assays of HIF-1α antibodies and the

predicted binding sites of the Netrin-1 promoter using WATs extracted from mice that were maintained on normal chow or HFD. **F** Results of ChIP-qPCR analysis of the Netrin-1 promoter and HIF-1α antibody binding using WATs extracted from mice that were maintained on normal chow or HFD, $n = 3$. Data are shown as mean ± s.e.m. *$P < 0.05$, **$P < 0.01$ over the control group as analyzed by unpaired t-test with Welch's correction.

---

hypoxia resulted in elevated HIF-1α expression, which contribute to adipose tissue fibrosis and encumbered the proliferation of preadipocytes[32]. Previous studies also reported that high-fat feeding-induced HIF-1α upregulation could inhibit adipogenesis by attenuating PPARγ expression as well as its activity[12]. Moreover, reports have also shown that hypoxia induced Netrin-1 expression in different tissue types[33,34]. Thus, CoCl₂ exposure led to upregulation of Netrin-1 and its receptor UNC5b expression in macrophages within the atherosclerotic plaques in a HIF-1α-dependent manner[34]. Myeloid-organed HIF-1α provided cardio protection by inducing

neutrophil Netrin-1 activity during ischemic and reperfusion injury[33]. Increased production of myeloid-derived Netrin-1 has also been shown in lipopolysaccharide-exaggerated lung injury in response to HIF-1α stimulation[33]. As a result, to investigate the potential association between adipose hypoxia and the activity of adipose Netrin-1, we separated the WATs from healthy mice, and the extracted WATs were treated with CoCl₂ to induce hypoxia and HIF-1α upregulation. Netrin-1 expression was observed on both protein and mRNA levels in the CoCl₂-exposed WAT, accompanied by HIF-1α accumulation (Fig. 6A, B). Similar results were also

collected in 3T3-L1 cells treated with DMOG (another HIF-1α stabilizer) (Fig. 6C), both of which demonstrate association between HIF-1α and Netrin-1 in the adipocytes as well as preadipocytes in vitro.

As a transcription factor, HIF-1α regulates the expression of various genes during tissue hypoxia, including the vascular endothelial growth factor and glucose transporter-1[35,36]. After confirming the association between adipose Netrin-1 and HIF-1α in vitro, we further investigated whether HIF-1α directly regulate Netrin-1 expression by analyzing the Netrin-1 promoter using BIOBASE and known HIF-1α binding sequences (Fig. 6D). Meanwhile, quantitative chromatin immunoprecipitation (qChIP) analysis was also performed by using WATs extracted from healthy Flox mice that were maintained on high-fat diet or normal chow diet for 8 weeks. As shown in Fig. 4E, F, high-fat feeding significantly facilitated HIF-1α binding than normal chow diet, and the HIF-1α-Netrin-1 binding under high-fat feeding condition is likely achieved via the HBS-1 motif. Our data so far suggest that HIF-1α could directly bind to the Netrin-1 promoter, preferably via the HBS-1 binding site, and regulate adipose Netrin-1 expression during high-fat feeding-induced hypoxia.

## Discussion

Systemic insulin resistance has been shown to often associate with increased WAT mass and adipose dysfunction. On the other hand, adipose dysfunction, such as insufficient adipogenesis and inappropriate energy storage, would in turn impair insulin sensitivity[37]. In order to cope with the milieu of metabolic changes, the adipose tissue would sometimes undergo transformations or so-called adipose tissue remodeling. Healthy adipose tissue remodeling generally includes enhanced adipogenesis, preferential subcutaneous fat expansion, while pathological adipose tissue remodeling often incurs impeded adipogenesis, tissue fibrosis and preferential visceral fat accumulation[4]. When facing excessive calorie intake, as often happens in modern society, the manner of adipose tissue remodeling matters significantly in maintaining the physiological metabolic equilibrium[8].

Netrin-1 is typically known as a neuron guidance molecule. We have, however, inadvertently observed difference of serum Netrin-1 level between diabetic and healthy individuals. Further analysis based on data obtained from the GEO databases also revealed upregulation of Netrin-1 in obese individuals and T2D subjects when compared to healthy controls, implicating a potential metabolic impact of Netrin-1 in the adipose tissue. Indeed, a few previous studies have reported a detrimental impact of circulating Netrin-1 on insulin resistance by exacerbating adipose inflammation, particularly the adipose tissue macrophage function. However, considering that the exact effects and mechanisms of adipose Netrin-1 on metabolic regulation remain unclear, we set to investigate the impact of adipose Netrin-1 on adipose activity and systemic metabolism.

By using an adipose Netrin-1 knockout mouse model (Ntn1^AKO), it was observed that under high-fat feeding condition, adipose Netrin-1 deletion improved metabolic parameters, including improved glucose tolerance and insulin sensitivity, reduced levels of fasting plasma glucose, triglyceride and free fatty acid. These observations are consistent with previous study also reporting improved glucose and insulin tolerance as well as lowered fasting plasma glucose, insulin and FFA levels in HFD-fed C57BL/6 mice that received transplantation of Netrin-1-deficient bone marrows[12]. Although mice reported in that earlier study had been maintained on HFD for 20 weeks following bone marrow transplantation, significantly longer than the 8-week HFD feed period used in the current study. Adipose Netrin-1 deletion also led to preferential accumulation of the iWATs, resulting in a metabolically "healthy" phenotype[38]. These are contrary to data presented in the previous study observing no changed or reduction of body weight/fat mass from mice with macrophages lacking Netrin-1[12], indicative of a significant effect of non-monocyte-origined, primarily adipose-derived Netrin-1 on body weight and fat mass. Moreover, these data also suggest that changes in the systemic metabolic parameters, including insulin sensitivity, glucose tolerance not only can be caused by deficiency of macrophage Netrin-1 per se, but can also be the consequence of adipose Netrin-1 deficiency.

Similar to earlier study reporting distinctly different Netrin-1 expression patterns between the preadipocytes and mature adipocytes in obese individuals[39], we also observed accumulation of Netrin-1 in mouse preadipocytes and less so in the mature adipocytes, implicating an arresting role of Netrin-1 on preadipocyte lipogenic differentiation. Subsequent mechanistic investigation demonstrated that the Netrin-1-elicited inhibition on adipogenesis is achieved via activation of the Wnt/β-catenin signaling pathway. Attenuated adipogenic PPARγ signaling cascade was also identified. In addition to the adipogenic effect of PPARγ activity per se, Netrin-1-dependent PPARγ inhibition may contribute toward the anti-adipogenic effect of Netrin-1 in preadipocytes, in line with previous studies reporting effects the Wnt/β-catenin on PPARγ and preadipocyte differentiation[29]. Based on these observations, we hypothesize that Netrin-1 secreted by mature adipocytes may affect the surrounding preadipocytes in a paracrine manner, activating the Wnt/β-catenin signaling pathway to suppress adipogenesis (Fig. 5). Indeed, there have been reports showing that β-catenin in mature WAT adipocytes were able to induce preadipocytes proliferation[30], although in the present study, the level of β-catenin was found to be unaffected by adipose Netrin-1 deletion in the Ntn1^AKO mice. Considering the pivotal and multifaceted properties of the Wnt/β-catenin signaling pathway, several components along the Wnt/β-catenin cascade have all been shown to be associated with inhibition of lipogenic differentiation[40–42]. As a result, how Netrin-1 interact with specific components of the Wnt/β-catenin cascade would worth further investigation.

In addition, here we observed elevated Netrin-1 level in undifferentiated preadipocytes, which gradually decreased during adipogenic differentiation. Though it may appear contradictory to Netrin-1 upregulation that was observed in mature primary adipocytes from high-fat fed mice[12], it is possible that other factors are also involved, which may influence adipose Netrin-1 expression. As subsequently shown, WAT Netrin-1 expression could be directly regulated by the hypoxic transcription factor HIF-1α under high-fat feeding condition. Since HIF-1α also suppresses lipogenic differentiation, our data again support an Netrin-1-mediated inhibition of adipogenesis in response to high-fat feeding-induced adipose hypoxia[19,20]. Moreover, by overexpression of Netrin-1 in the WATs via AAV in high-fat-fed mice, we again observed a detrimental role of adipose Netrin-1 on glucose homeostasis, accompanied by limited adipogenesis.

There are, of course, limitations regarding the present study. The receptors involved in mediating the role of Netrin-1 in the adipose tissues would require detailed future investigation[43], via which multiple signaling pathways downstream of Netrin-1 may be elucidated. Moreover, as already mentioned in the Results section, non-adipose expression has been reported of the aP2-cre mice[44], although expression of fabp4/aP2 in adipose macrophage and livers are low[21], suggesting limited participation of adipose macrophage-derived and hepatic Netrin-1 on the metabolic effects observed from the Ntn1^AKO mice. Indeed, the metabolic phenotype reported here is significantly different to the previous study[12], in which the average body weight and fat mass of the HFD-maintained mice with/without Netrin-1-deficient bone marrow transplantation were unchanged, in contrast to a significant difference in body weight and fat mass between the Ntn1^AKO and Flox mice (Fig. 1J–N). In addition, studies focusing on the metabolic influence of other macrophage-derived effectors also showed no change in adipose mass or body weight[45,46], confirming a lesser role in regulating adiposity and lipid storage of the recruited hematopoietic stem cell-derived macrophages, which were better established to be associated with obesity, inflammation and insulin resistance[45]. Given the above, while also considering the likely impact of Netrin-1 on preadipocyte differentiation, the preadipocyte knockout Pdgfra-Cre mice could be implemented for further exploration. Concurrently, gender differences frequently lead to divergent metabolic phenotypes under high-fat diet conditions. Notably, male mice exhibit heightened susceptibility to diet-induced glucose dysregulation compared to female counterparts[47]. In our investigation, the exclusive utilization of male subjects imposes limitations on the generalizability of our findings. Subsequent research will examine whether gender difference

**Fig. 7 | Schematic representation of hypothesized mechanisms of Netrin-1 in the white adipose tissue during HFD-induced hypoxia.** HFD-induced adipose hypoxia upregulates Netrin-1 expression in the mature adipocytes. This increased level of Netrin-1 would exert a paracrine effect on local preadipocytes, affecting preadipocytes proliferation and function. Thus, Netrin-1 would disrupt adipogenesis and fat storage, an important compensatory approach during excessive calories intake, by inhibiting PPARγ activity. The inhibition of PPARγ activity could possibly be conducted through the Wnt/β-catenin signaling pathway, which has been reported to inhibit the PPARγ activity, resulting in impeded adipogenesis and abnormal fat deposition. Graphic created in BioRender. Winx, S. (2026) https://BioRender.com/8zntys0.

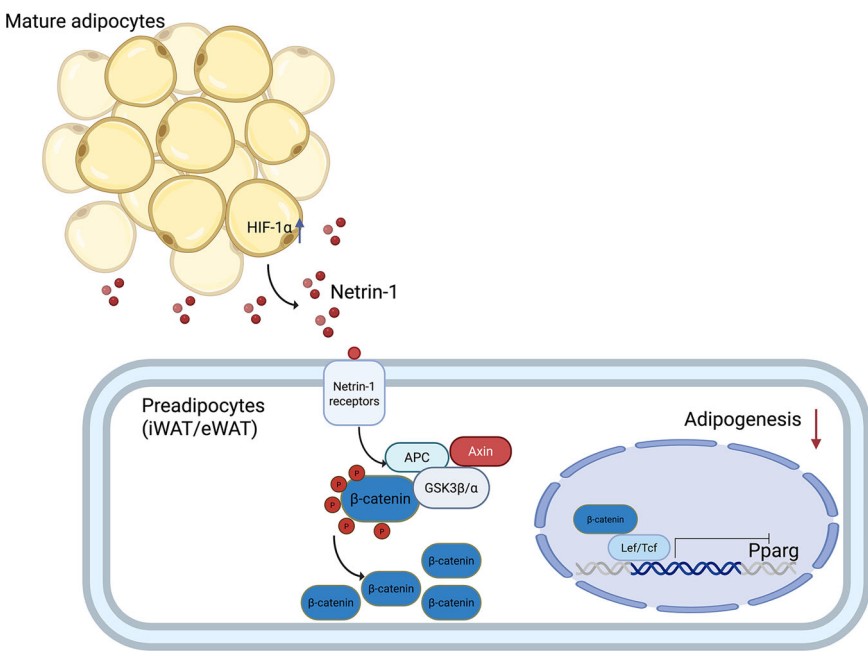

modulates Netrin-1 expression or function in the adipose tissues, thereby influencing the systemic metabolic outcome.

In summary, we have shown in the present study that adipose Netrin-1 expression could be mediated by environmental factors in response to metabolic changes. For instance, high-fat feeding could induce adipose dysfunction such as adipose hypoxia. The resultant activation of HIF-1α would then facilitate upregulation of mature adipocyte-derived Netrin-1. Adipose Netrin-1 in turn inhibit the WAT remodeling processes, particularly preadipocytes differentiation, by attenuation of the PPARγ signaling and stimulation of the Wnt/β-catenin signaling pathways, both of which result in deregulated adipogenesis and exacerbated systemic glucose tolerance and insulin sensitivity (Fig. 7).

The present study may provide new insights into therapeutic development for HFD-induced obesity and T2D by revealing additional peripheral roles of the traditionally known neuronal factors. Moreover, since neuronal input is crucial in metabolic regulation, establishing neural-metabolic axis based on the impact of locally derived neuronal factors in metabolic organs would also contribute to our current investigation regarding energy expenditure and metabolism.

## Materials and methods
### Mature adipocytes and stromal vascular fractions (SVFs) cell isolation
Mice were euthanized by cervical dislocation performed by trained personnel, with death confirmed by the absence of vital signs and reflexes. Primary SVFs and mature adipocytes were isolated from the inguinal WATs (iWATs) of 4-week-old male mice as described previously[37]. Briefly, mouse WATs were extracted, rinsed three times in PBS supplemented with 1% penicillin/streptomycin (vol./vol.), then minced and digested with 1 mg/mL type 2 collagenase (Sigma-Aldrich, C6885) that was pre-dissolved in digestion buffer (0.4 g/mL KCl, 0.06 g/mL $KH_2PO_4$, 8 g/mL NaCl, 0.09 g/mL $Na_2HPO_4$, 1 g/mL glucose, 1.2 mmol/L $CaCl_2$, 1 mmol/L $MgCl_2$, 0.8 mmol/L $ZnCl_2$, 3% [vol./vol.] bovine serum albumin [BSA]) for one hour at 37 °C. The mixture was then filtered through a 70 µm filter (Falcon, 352350) and centrifuged at $500 \times g$ for 10 min to collect the preadipocytes and mature adipocytes in suspension.

### Cell culture and differentiation
The undifferentiated mouse preadipocyte cell line, 3T3L-1, obtained from ATCC (ATCC Number CL-173), was maintained in cell culture medium (DMEM [Gibco, 11965092] supplemented with 10% [vol./vol.] heat-inactivated newborn calf serum [NBCS], 1% [vol./vol.] penicillin/streptomycin, 1% [vol./vol.] L-glutamine and 1 mmol/L sodium pyruvate). For induction of adipocyte differentiation, 2 days after confluence, the 3T3L-1 cells were induced to differentiate by adding DMEM containing 10% FBS and 1 µg/mL insulin (Sigma-Aldrich, I6634), 0.5 mmol/L 1-methyl-3-iso-butyl-xanthine (IBMX) (Sigma-Aldrich, I7018) and 0.25 µmol/L dexamethasone (Sigma-Aldrich, D4902). The differentiation-inducing media were replaced with DMEM supplemented with 10% FBS after 2 days, and the medium was changed every 2 days. Mice were euthanized by cervical dislocation, and the extracted SVFs isolated from the WATs were cultured with DMEM/F12 (Gibco, 11320033) containing 10% NBCS and 1% penicillin/streptomycin. Differentiation induction was initiated by adding DMEM/F12 containing 10% fetal bovine serum (FBS) and 0.5 µg/mL insulin (Sigma-Aldrich, I6634), 0.5 mmol/L IBMX (Sigma-Aldrich, I7018), 1 µmol/L dexamethasone (Sigma-Aldrich, D4902). The induction media were then replaced with DMEM/F12 containing 10% FBS, 1% penicillin/streptomycin, and 0.5 µg/mL insulin and the media were then changed every other day. Isolated mature adipocytes were seeded into 6-well plates at approximately 1000 µL cell suspension per well. Cells were maintained in basal media consisting of DMEM/F12 medium containing 5 mmol/L glucose, supplemented with 10% FBS, 50 µg/mL gentamicin, 1% amphotericin B, and 1% penicillin/streptomycin. Cells were tested for mycoplasma contamination using the MycoBlue™ Mycoplasma Detector kit (Vazyme, Cat# D101-01). All cells were verified free of mycoplasma prior to experimental use.

### AAV plasmid construction
For in vivo observation of adipose Netrin-1 overexpression, mouse Netrin-1 (NM_008744.2) was obtained from the cDNA library of Genechem (Shanghai, China) with the following primer sequences: Ntn1 forward: 5′-GGAGGTAGTGGAATACCGGTCGCCACCATGATGCGCGCTGTGTGGGGAG-3′ and reverse: 5′-ACCATGGTGGCGGGGATCCACGGCCTTCTTGCACTTGCCCTTCTTC-3′.

The AAV9 vector plasmid GV585 (FABP4p-MCS-EGFP-3Flag-SV40 PolyA, from Shanghai Genechem Co., Ltd.) and the Ntn1 gene sequence were digested by restriction enzymes AgeI and BamHI, and complete cloning was carried out by the in-fusion recombination method. The recombinant vector was detected by DNA sequencing. The aP2 promoter serotype 9 AAV vector was then selected since it has been reported that it could promote efficient gene transfection in WATs[44]. The control-aP2-AAV9 was generated in the same process by using the empty GV585 vector (FABP4p-MCS-EGFP-3Flag-SV40 PolyA, GeneChem).

## Adeno-associated virus production

The viral vectors were transfected into the 293T cells using Lipofectamine 2000 (Invitrogen; Thermo Fisher Scientific, Inc.) together with plasmids pHelper and pRepCap. The AAVs were harvested 72 h post-transfection, and the AAV9 were purified through iodixanol gradient ultracentrifuge. Purified AAV viruses were titered using a quantitative PCR-based method. All AAVs used in this study were prepared in 0.001% Pluronic F-68 solution (Poloxamer 188 Solution, PFL01-100ML, Caisson Laboratories, Smithfield, UT, USA).

## Animals

All animal experiments were approved by the Institute of Radiation Medicine, Chinese Academy of Medical Sciences Animal Care and Use Committee (IRM-DWLL-2019019). We have complied with all relevant ethical regulations for animal use. All mice used in the present project were male. Mice were housed in a temperature-controlled ($23 \pm 2\,°C$) environment that were kept at a 12 h light-dark cycle. All animals were maintained under strict inbreeding conditions in pathogen-free individually ventilated cages within our accredited animal facility. The animals were housed as 3–5 mice per cage. Mice were randomly assigned to the experimental groups, with six mice per group. For the present study, 8-week-old male mice were maintained on a normal chow diet (1010009, Xietong, Jiangsu, China) or HFD (H10060, Huafukang Bioscience, Beijing, China) *ad libitum* with free access to water.

The *Ntn1*[AKO] mice were generated by the Shanghai Model Organisms Center Inc. Briefly, to obtain the *Ntn1*[AKO] mice, the C57BL/6Smoc-*Ntn1*[em1(flox)Smoc] mice (NM-CKO-210064) and the *aP2-cre* mice: B6.Cg-Tg(*Fabp4*-cre)Smoc (NMX-TG-192032) were selected for intercross breeding. A two-step breeding strategy was employed. First, Flox/+ males and females were intercrossed to generate Flox/Flox mice. Concurrently, Flox/+ mice were crossed with aP2-Cre mice to yield Flox/+; *aP2*-Cre mice. Subsequently, Flox/Flox mice were crossed with Flox/+; *aP2*-Cre mice to obtain Flox/Flox; *aP2*-Cre mice. The Cre allele was inherited paternally. The genotype was identified by PCR with the following sequences: Flox-forward: TAAGGGAGCAAGGAGAATGGG, Flox-reverse: TCGGTGGAGTGGC GGGCAAG, Cre-forward: TCGATGCAACGAGTGATGAG, Cre-reverse: TCCATGAGTGAACGAACCTG. The background strain of both the *Ntn1*[fl/fl] and the aP2-cre mice was C57BL/6J, as provided by the Shanghai Model Organisms Center Inc. Netrin-1 floxed littermates were used as controls. The aP2-promoter affects all types of adipose tissues, including the BATs and WATs[21,44]. In addition, 6–8-week-old male C57BL/6J mice (219, CSTR:15497.09. N000010) were purchased from Beijing Vital River Laboratory Animal Technology Co., Ltd. Mice were allowed to get used to the environment for at least 1 week before experiments. For Netrin-1 overexpression, 0.1 mL of the aP2-AAV9-Ntn1 or Control-aP2-AAV9 (total of 3E + 11 v.g., diluted in PBS) was injected into the tail vein of each mice (8-week-old, male, C57BL/6J).

## Gene expression analysis

For undifferentiated 3T3-L1 preadipocytes and cells from the SVFs, cells were seeded and maintained in 6-cm dishes. When the cells reached 70% confluence, they were infected with either GFP-Ntn1 lentivirus or control virus. The cell media were replaced after 24 h, and transduction efficiency was assessed after 48 h by immunoblotting and qPCR.

For differentiated 3T3-L1 adipocytes, cells were infected with GFP-Ntn1 lentivirus or control virus on day 6 following differentiation induction. Overexpression efficiency was then evaluated by immunoblotting and qPCR.

Following stable overexpression of Netrin-1 in cells from the SVFs, we supplemented the adipogenic differentiation medium with 2 μM rosiglitazone to perform the phenotypic rescue experiment.

## Glucose tolerance test (GTT) and insulin tolerance test (ITT)

For GTTs, mice were fasted for approximately 8–10 h (overnight). A 10% (w/v) glucose solution was prepared by dissolving D-glucose (Solarbio, G8151) in sterile physiological saline, followed by sterilization through a 0.22 μm membrane filter (Millipore, SLGPR33RB). The glucose solution was then injected intraperitoneally into each mice at a concentration of 1 g/kg body weight. Blood glucose levels were monitored with a glucose meter (OneTouch® Ultra™, Johnson) at 0, 15, 30, 60, and 120 min after injection by measuring blood drawn from a small incision of the mouse tail vein at each time point. For ITTs, recombinant human insulin (Sigma, I9278) was diluted to 0.075 IU/mL in sterile insulin diluent, followed by sterilization through a 0.22 μm membrane filter (Millipore, SLGPR33RB). Mice were fasted for approximately 5 h before intraperitoneally injection of insulin solution at 0.75 U/kg body weight. The blood glucose levels were measured monitored with a glucose meter at 0, 15, 30, 60, and 120 min after injection the same as for the GTTs. The *Mus musculus* C57BL/6J strain was utilized for animal experiments.

## Histology and adipocyte size analysis

Mice were euthanized by cervical dislocation performed by trained personnel, with death confirmed by the absence of vital signs and reflexes. Mouse tissue samples were extracted and fixed for 12–24 h in 5–10% paraformaldehyde before being embedded in paraffin. Tissues were then sectioned using a Leica Biosystems' rotary microtome before subsequent H&E and Sirius Red staining. Adipocyte areas were manually measured using ImageJ software with ≥100 adipocytes analyzed per animal. Adipocyte size frequency distributions were determined according to previous studies[48]. Briefly, objects with areas <350 μm² were excluded as potential mixtures of adipocytes and stromal vascular cells. Frequencies were computed using the FREQUENCY function (=FREQUENCY (data_array, bins_array)) in Microsoft Excel with bin intervals of 1000 μm². Total adipocytes within the distribution were then quantified to convert frequencies into percentages of counted adipocytes.

## Measurement of systemic metabolic parameters

The body weights of mice from all experimental groups were measured and recorded weekly. Mouse body composition was evaluated by magnetic resonance imaging (MRI, EchoMRI™-100H) to examine lean and fat mass distribution. Littermate-matched experimental and control mice (same gender and age) were fasted for 6–8 h. Following successful system calibration with the dedicated rapeseed oil tube, murine body composition analysis is performed by placing the subject in an appropriately sized crimson tube holder corresponding to its mass range. The mice were gently advanced to the distal end using an inner pusher probe for immobilization, whereupon the loaded assembly was fully seated into the instrument's lower aperture. The scanning was initiated via the software interface using the mice/primary accumulation/parameter 1 protocol, with essential metadata, including animal ID and body weight, recorded in the dialog box triggered by the Start Scan command. Male 8-week-old mice were also housed individually in separate metabolic cages to record physical activity, food intake, oxygen consumption, and carbon dioxide production (TSE Systems, Germany). After that, the mice were euthanized by cervical dislocation, with death confirmed by the absence of vital signs and reflexes.

## Western blot analysis

Total proteins were extracted from the mouse tissues, primary adipocytes, and 3T3-L1 cells using the TRIzol-procotol. To examine the protein expression levels of the cell signaling elements, mice were euthanized by cervical dislocation and mature adipocytes isolated from *Ntn1*[AKO] and Flox mice were homogenized with an appropriate volume of RIPA buffer (Beyotime, China) containing protease and phosphatase inhibitors (Thermo Scientific,78446). Electrophoresis was performed, and the separated protein samples were transferred to the PVDF membranes (Millipore, USA). After blocking with 5% (wt/vol.) BSA (Beyotime, China), the membranes were incubated with primary antibodies raised against Netrin-1 (rabbit origin, 1:1000; Ab126729, Abcam), HIF-1α (rabbit origin, 1:1000; 36196, CST), GSK3β (rabbit origin, 1:1000; A11731, ABclonal), P-GSK3β (rabbit origin, 1:1000; 9931S, CST), β-catenin (rabbit origin, 1:1000; A19657, ABclonal), active β-catenin (rabbit origin, 1:1000; 8814, CST), Rpl26 (rabbit origin, 1:1000; A9192, ABclonal), actin (rabbit origin, 1:5000;

AC026, ABclonal), PPARγ (rabbit origin, 1:1000; A0270, ABclonal), pan-AKT (rabbit origin, A18120, 1:1000, ABclonal), phospho-AKT (rabbit origin, AP1208, 1:1000, ABclonal) at 4 °C overnight. The PVDF membranes were then incubated with secondary antibodies (rabbit origin, AS014, 1:5000, ABclonal) for 1 h at room temperature before visualization by adding the ECL buffer (Merck Millipore, WBKLS0100) on an AI600 imager (Thermo Fisher Scientific, USA).

### ChIP analysis

The epididymal WATs (eWATs) and iWATs from a total of 6 different mice were combined per run. Mice were euthanized by cervical dislocation, and all WATs were chopped and fixed with 1% methyl alcohol for 20 min at 37 °C and incubated with 125 mM glycine for 10 min before lysed with a ChIP lysis buffer (1% SDS, 5 mmol/L EDTA, 50 mmol/L Tris-HCl pH8.1). Tissue lysates were further fragmented by ultrasound and diluted as 1:10 with a dilution buffer (1% SDS, 0.1 NaHCO$_3$) and used for immunoprecipitation with antibodies against HIF-1α (rabbit origin, 1:100; 36196, CST) and IgG (rabbit origin, 1:100; 2729, CST) at 4 °C overnight. The immunoprecipitation products were then incubated with protein G (Invitrogen, 10003D) at 4 °C for 4 h the following day. The beads were washed with ChIP buffer I (0.1% SDS, 1% Triton X-100, 2 mmol/L EDTA, 20 mmol/L Tris-HCl pH 8.1, 150 mmol/L NaCl), ChIP buffer II (0.1% SDS, 1% Triton X-100, 2 mmol/L EDTA, 20 mmol/L Tris-HCl pH8.1, 500 mmol/L NaCl), ChIP buffer III (0.25 mol/L LiCl, 1% NP-40, 1% deoxycholate, 1 mmol/L EDTA, 10 mmol/L Tris-HCl pH 8.1) and TE buffer (10 mmol/L Tris-HCl pH 8.1, 1 mmol/L EDTA). The beads were then eluted with the dilution buffer at 37 °C for 2 h before being treated overnight at 65 °C for crosslinking reversion. The DNA products were cleaned up using a PCR cleanup Kit (Macherey-Nagel, Germany). ChIP-qPCR was then performed using a TransStart TopGreen qPCR Supermix (TransGen Biotech), and quantitative real-time PCR was performed on an ABI 7500 System.

### RNA extraction and real-time PCR

Total RNAs of the adipose tissues or adipocytes were extracted using the TRIzol method (Invitrogen). Reverse transcription was carried out using the ABScript II cDNA First-strand synthesis kit (ABclonal) to generate cDNAs. Real-time PCR amplification was then performed using an SupRealQ Ultra Hunter SYBR qPCR Master Mix(U+) (Vazyme Biotech Co., Ltd) on an ABI 7500 System (primer sequences were listed in the Supplementary Table 1).

### RNA-seq

Total RNAs were isolated using a TRIzol total RNA extraction kit (TIANGEN, Cat.No. DP424), which yielded > 2 μg of total RNAs per sample. The RNA quality was examined by gel electrophoresis on 0.8% agarose and spectrophotometry. High-quality RNA with a 260/280 absorbance ratio of 1.8–2.2 was used for library construction and sequencing. Illumina library construction was performed according to the manufacturer's instructions (Illumina, USA). Oligo-dT primers are used to transverse the mRNAs to obtain cDNAs (APExBIO, Cat. No. K1071). The second chains of cDNAs were amplified and purified by an AMPure XP system (Beckman Coulter, Beverly, USA). After library construction, library fragments were enriched by PCR amplification and selected according to a fragment size of 350–550 bp. The library was quality-assessed using an Agilent 2100 Bioanalyzer (Agilent, USA). The library was sequenced using the Illumina NovaSeq 6000 sequencing platform (Paired end150) to generate raw reads. RNA-seq data are available for downloading from the GEO database with the accession number GSE254894.

### Serum analysis

After a 16-h fast, tail vein blood was collected via sterile lancet puncture with the first 1–2 drops discarded. Samples were drawn into heparinized capillary tubes, wounds cauterized immediately, and the collected blood samples were centrifuged at 4 °C to obtain serum samples. Serum triglyceride and free fatty acid levels were determined using a LabAssayTM triglyceride assay kit (632-50991, Wako Chemicals, USA) and a free fatty acid (FFA) assay kit (MAK466, Sigma-Aldrich, USA).

### Statistics and reproducibility

All data sets were subjected to statistical tests (Shapiro–Wilk or Kolmogorov–Smirnov) for normal distribution assessment using the SPSS software (SPSS27) before further statistical analyses. Quantitative data are expressed as mean ± standard error of mean (s.e.m.) of at least three independent experiments. Unpaired t-test with Welch's correction was applied for difference analysis between two groups, and one-way ANOVA was used for comparisons between multiple groups. All statistical tests were analysed by the GraphPad Prism 10 software. Statistical significance was defined as $P < 0.05$.

To ensure reproducibility, each experiment includes at least three independent replicates. To ensure consistency, the replicate number and sample size for each experiment are detailed in the "Materials and Methods". To ensure experimental reproducibility, all replicates were performed under identical conditions.

## Data availability

All data generated and analyzed during this research are available as follows: RNA-seq datasets have been deposited in the Gene Expression Omnibus (GEO) under accession GSE254894; Source data for graphical representations in the main figures are provided as Supplementary Data; any additional information, including complete raw datasets, is accessible from the corresponding author upon reasonable and justified request.

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

## Acknowledgements

We are grateful for the generous financial support of the National Key Research and Development Programme of China (no. 2020YFA0803700), the Tianjin Natural Science Foundation Project (no. 25JCZDJC00270), and the National Natural Science Foundation of China (no. 81760327). We also thank the APExBIO Technology LLC (Shanghai, China) for the RNA-seq service and subsequent bioinformatics analysis.

## Author contributions

Li C. is the guarantor and takes full responsibility for the work as a whole. Mao D. and Kong D. supervised the experimental design and reviewed the manuscript. Shi H., Tang J., Yan X., Ke T. conducted the experiments and analyzed data.

## Competing interests

The authors declare no competing interests.

## Additional information

**Supplementary information** The online version contains Supplementary material available at https://doi.org/10.1038/s42003-026-09749-x.

