## [Transparent Peer Review File · Communications Biology]

Netrin-1 disrupt high-fat-diet-induced adipogenesis via the PPAR γ and Wnt/ β -catenin signaling pathways

Corresponding Author: Professor Chen Li

Version 0:

Reviewer comments:

Reviewer #1

(Remarks to the Author)

GENERAL COMMENTS:

This manuscript contains multiple animal, cell and molecular experiments on the adipose role of netrin-1, starting from bioinformatic examination of existing human publicly available data presumably leading to the authors' hypothesised role for netrin-1 in adipose tissue to regulate adipogenesis, mainly in subcutaneous fat depots. This is in principle a well-structured study with interesting findings that are important to be communicated, however the manuscript in its current form requires substantial revision. The structure of the manuscript is problematic and confusing, methods are inadequately described and many of the results rely on non-quantified western blot data. All of this should be rewritten / reanalysed before it can be accepted.

The animal experiments are often carried out using either normal chow diet or high fat diet, and I was concerned that this is not a correct control for HFD, however the authors have not made any formal comparison between diets. This is okay, but does limit interpretation of results and authors need to be careful not to make these comparisons in their conclusions /discussion.

Interestingly, expansion of adipose tissue (of those measured) appears to have been seen mainly in inguinal (subcutaneous) depots, considered a more "safe" lipid storage depot. This comes with the caveat that only a selection of adipose depots have been measured, and the use of eWAT is controversial as a true visceral WAT depot at least when modelling human disease, as this tissue does not exist in people.

It appears that the identification of Netrin-1 as the hypothesised regulator of metabolic phenotype was carried out through bioinformatic analysis of publicly available data, but this is not mentioned at all in the introduction, where discussion of macrophage effects are mentioned, and nor in the methods, appearing for the first time in the start of the results section. This should be addressed in the structure of the manuscript as it makes the narrative disjointed and confusing.

SPECIFIC COMMENTS:

INTRODUCTION

Line 90 "diabetic individuals" should be "people living with diabetes" throughout.

Line 97: What do the authors mean by "the ways via which white adiposes (WATs) expand" – please be more specific and expand on this.

Line 121 from "Thus, netrin-1..." check grammar of this sentence.

Line 128 "plagues" should be "plaques"

Line 138: mention of macrophages, almost in passing. Please expand on this, if relevant to the present study, or remove, as it is confusing why this is mentioned here but not at all investigated in this manuscript.

METHODS

The methods are largely insufficient. Whole techniques are not mentioned – e.g. bioinformatic analysis, cell culture stable over expression, rosiglitazone experiment, primary adipocyte experiment, adipocyte size analysis, recombinant netrin-1 etc. All animal experiments should be described in detail according to ARRIVE guidelines, and not just in general. GTT/ITT methods are not described sufficiently, all mouse strains should be clearly defined for every experiment. Is it C57BL6/J or N? qPCR primer sequences are not listed.

Crucially, the Ap2 promotor used for this CRE and AAV overexpression experiments is not defined, and the authors mention towards the end of the paper that this is found in ALL adipose tissues and therefore not exclusive to WAT. This should at minimum be defined in the methods, with a clear indication of what AP2 is / where it is found. Could the authors either

reference or demonstrate the localization of this cre eg. by cross to reporter mouse and subsequent staining of tissues?
Line 202: remove "distribution".

RESULTS

The Results section, is very clearly written as a "results and discussion", with large sections that should be moved to the discussion section of this article:

e.g. Lines 325-328, 334-340, 346-356, 392-394, 447-450, 453-460, 474-476,

None of the western blot results are quantified, with only letterbox example blots shown throughout. Please quantify all of these data. Without this, none of these results / conclusions can be accepted and it is very difficult to discuss further.

Line 421 from "immunoblotting": This blot is very variable for control samples and without quantification cannot properly accept this..

Line 429: not quantified - this panel is not enough to convince of this.

Line 490 -492 from "the ap2.." this should be in the methods section.

Line 528: this crucial information should have been mentioned in the methods.

The structure of the results section is confusing, going backwards and forwards between different mouse and cell models. Suggest to restructure, with all animal experiments coming before molecular / cell experiments, and showing all the AKO data first before over expression then cell data.

The over-expression experiment is nice because the findings are in opposition to the KO model, thus supporting the role they initially describe.

DISCUSSION

This feels largely a repeat of the results key findings and some discussion already mentioned should appear here and be more concise.

Line 560 "inadvertently" – what do the authors mean? Is there a reference missing here?

Line 574-575 from "Adipose": Please expand on this sentence and justify this reference

Line 611: Inappropriate citation.

Line 612 onwards: Mention of macrophages, please could the authors elaborate this link to macrophages more clearly as it isn't clear to this reviewer what the context is.

Line 629 and in the abstract: Authors refer to "environmental factors" but should only state high fat diet here. Although, because the HFD vs control has not been formally compared in any experiment in this manuscript, they can't really conclude that either. In order to make that conclusion authors would have to repeat experiments using a nutrient matched control diet (not standard chow) and make statistical comparison.

Reviewer #2

(Remarks to the Author)

In this manuscript, Shi et al investigated the implications of Netrin-1 elevation in circulation in obesity and diabetes. The authors used newly generated adipocyte specific Netrin-1 KO mouse model to explore the in vivo role of Netrin-1. However, this study does not provide sufficiently convincing results to substantiate the current schematic model, limiting its potential impact.

1. Figure 1 provides the only piece of evidence supporting adipocytes as a producer of Netrin-1, with poor quality of western blot, no quantification, and no validation with other methods on whether adipocytes are a key source of elevated serum Netrin-1 in obese and diabetic patients. After the generation of aKO mice, there were no data showing the contribution of adipocytes to its tissue and systemic levels.
2. The authors argued that the improved insulin sensitivity may be explained by the increase in adipogenesis in inguinal but not other fat depots. There may be other possibilities such as altered adipose tissue secretome.
3. The inhibitory effects of Netrin-1 on adipogenesis were suggested selective in inguinal fat. While the study using 3T3-L1 adipocytes as well as downstream targets PPAR γ and Wnt are well studied for white adipocytes in general.
4. In obesity, both visceral and inguinal fat expand with increased adipogenesis, which was not consistent with the summary in figure 7.
5. Visceral fat even experiences higher levels of hypoxia than inguinal fat in obesity. Whether induced HIF α leads to impaired adipogenesis of inguinal fat through Netrin-1-mediated interaction between adipocyte and preadipocyte under this condition requires further clarification.

Reviewer #3

(Remarks to the Author)

Shi et al. investigate the role of Netrin-1 in the development of white adipose tissue (WAT) expansion and glucose intolerance in obesity. They find that adipocyte-specific Netrin-1 knockout mice display increased WAT expansion and improved glucose metabolism, whereas AAV-mediated Netrin-1 overexpression impairs glucose tolerance and reduces PPAR γ levels in WAT. In vitro, Netrin-1 reduced PPAR γ expression thereby blunting adipogenesis. Overall, authors conclude that obesity-induced Netrin-1 restrains WAT expansion due to blunted adipogenesis thereby increasing liver lipid accumulation and impairing glucose metabolism. This is an interesting paper using different mouse and cell culture models. However, to support their conclusions, several experiments need to be performed.

Major comments:

Figure 2: data from only very few HFD mice (n=4) are presented. Are these data coming from one cohort of mice? Did authors confirm these finding in at least two other independent cohorts of mice showing the same significance? If yes, please add these data to the manuscript. If not, this needs to be done since in vivo data gained with an n of 4 mice per group may be underpowered.

Figure 2Q: authors show only one H+E stained liver section to suggest that KO mice accumulate less lipids in livers. This is clearly not enough. Please quantify liver lipid content using a biochemical assay in livers of HFD-fed control and knockout mice. Similarly, please show pAkt from more than n=1 (Supplementary Figure 2) and quantify bands of Western blots to support the notion that adipocytes of KO mice show increased insulin signaling.

Figure 3: Authors show that Netrin-1 overexpression decreases ORO in primary adipocytes (Fig. 3G/H) and PPAR γ expression in 3T3-L1 adipocytes. Are PPAR γ levels also decreased in primary adipocytes? Moreover, treatment of 3T3-L1 preadipocytes with Netrin-1 promotes β -catenin and authors conclude that "our results demonstrate that adipose Netrin-1 mitigate adipogenic differentiation of preadipocytes by activating the β -catenin pathway". However, such link is not shown. Authors need to repeat experiment in Nestin-1 overexpressing preadipocytes and/or in recombinant Netrin-1-treated preadipocytes after inhibiting/knowing down β -catenin and show that inhibition of the β -catenin pathway blunts the negative effect of Netrin-1 on adipogenesis.

Figure 5: Please analyze and quantify PPAR γ protein levels from several mice per group (Fig. 5E). Based on the hypothesis, one would expect that HFD-fed overexpressing mice reveal reduced fat mass/mass of iWAT (i.e. reduces WAT expansion) as well as increased liver lipid accumulation (lipid spillover, ectopic fat accumulation). Therefore, these parameters (WAT mass, liver lipid levels) need to be shown.

Minor comments

Abstract, line 71: please adapt, since insulin resistance/sensitivity is not shown in overexpressing mice.

Supplementary Fig. 3E: quality of images is very poor and need to be improved. Please note that n=3 is shown in the graph but n=4 mice mentioned in the Figure legend. Please correct.

Supplementary Fig. 6: please check mRNA expression of Netrin-1 in livers similar as done for WAT

Line 325-331; 346-353: please move to Discussion

Please expand the discussion by adding the limitation of having only male (but not female mice) included in this study.

What is the background strain of Ntn1 $^{fl/fl}$ and Ap2-Cre mice? Why were Ap2-Cre rather than the more adipocyte-specific AdipoqCre-mice used in this study?

Statistics: where data checked for normal distribution and similar variance? If not, this needs to be done and statistical tests need to be adapted in case of non-normal distribution/non-similar variance.

Version 1:

Reviewer comments:

Reviewer #1

(Remarks to the Author)

This is the second time I have seen this (now revised) manuscript. All reviewer comments appear to have been adequately addressed, with clarifications added and additional data included in the manuscript to support the original findings

Reviewer #2

(Remarks to the Author)

The authors have experimentally addressed most of my concerns. I don't have additional comments.

Reviewer #3

(Remarks to the Author)

In the abstract, authors still mention that "...WAT Netrin-1 overexpression used in vivo adeno-associated virus delivery resulted in worsened insulin resistance in both high fat and normal chow-fed mice". This is not appropriate since no data looking at insulin sensitivity (e.g.. insulin tolerance test) are shown for overexpressing mice. Rather, a glucose tolerance test is presented, which is not only affected by insulin sensitivity but also insulin production. Please adapt the abstract accordingly.

Statistics: authors did not reply do my question whether data that were compared had similar variance. If variance between groups was not similar, this needs to be taken into consideration when doing t-tests (Welch's correction).

Please clearly state the background strain of used Ntn1 $^{fl/fl}$ and Ap2-Cre mice in the methods section.

Version 2:

Reviewer comments:

Reviewer #3

(Remarks to the Author)

Authors have adequately replied to my remaining open questions. I have no further comment.

Response to the Reviewers' comments:

Reviewer #1 (Remarks to the Author):

Line 97: What do the authors mean by “the ways via which white adiposes (WATs) expand” – please be more specific and expand on this.

Response: As per the reviewers' suggestions, the following content has been added in line 97 (Introduction) of the revised manuscript and highlighted in red:

Pathological adipose tissue expansion typically manifests as preferential accumulation of visceral adipose tissue, suppression of adipogenic differentiation, and dysfunctional adipocytes. Whereas metabolically healthy adipose expansion is characterized by preferential accumulation of subcutaneous adipose tissue and enhanced adipogenic differentiation capacity.

Line 121 from “Thus, netrin-1...” check grammar of this sentence.

Response: As per the reviewers' suggestions, the sentence has now been modified as the following. Changes are also highlighted in red in line 123 of the revised manuscript:

Indeed, Netrin-1 has been shown to participate in tumor initiation and progression by arresting malignant cell apoptosis. Tumor neoangiogenesis¹⁰, vascular diseases¹¹ and metabolism¹² are also facilitated by Netrin-1 as reported.

Line 128 “plagues” should be “plaques”

Response: We thank the reviewer for pointing this out; this spelling error has been corrected and highlighted in red in line 131 of the revised manuscript.

Line 138: mention of macrophages, almost in passing. Please expand on this, if relevant to the present study, or remove, as it is confusing why this is mentioned here but not at all investigated in this manuscript.

Response: After careful consideration and reflection on the reviewer's suggestion, we agree that the paragraph that described macrophages may indeed be confusing. We then decided to have this part removed.

The methods are largely insufficient. Whole techniques are not mentioned – e.g. bioinformatic analysis, cell culture stable over expression, rosiglitazone experiment, primary adipocyte experiment, adipocyte size analysis, recombinant netrin-1 etc.

Response: We appreciate the reviewer's suggestion and have included information regarding these techniques in the Methods section of the revised manuscript (highlighted in red).

All animal experiments should be described in detail according to ARRIVE guidelines, and not just in general. GTT/ITT methods are not described sufficiently, all mouse strains should be clearly defined for every experiment. Is it C57BL6/J or N? qPCR primer sequences are not listed.

Response: We have added experimental details regarding the GTT/ITT. Mouse strains used for animal studies were also specified. Sequences of all qPCR primers used in this

study have also been listed in ESM Table 1 of the Supplementary Material.

Crucially, the Ap2 promotor used for this CRE and AAV overexpression experiments is not defined, and the authors mention towards the end of the paper that this is found in ALL adipose tissues and therefore not exclusive to WAT. This should at minimum be defined in the methods, with a clear indication of what AP2 is / where it is found. Could the authors either reference or demonstrate the localization of this cre eg. by cross to reporter mouse and subsequent staining of tissues? Line 202: remove “distribution

Response: We appreciate the reviewer's insightful comments. The aP2 (FABP4) promoter is capable of driving transgene expression in both white adipose tissues (WATs) and brown adipose tissues (BATs), although its activity has been reported to be higher in the WATs than in the BATs (Jimenez V. et al., Diabetes. 2013,; Tang Y. et al., J Mol Cell Biol. 2016). The word “distribution” in line 202 has been removed.

The Results section, is very clearly written as a “results and discussion”, with large sections that should be moved to the discussion section of this article: e.g. Lines 325-328, 334-340, 346-356, 392-394, 447-450, 453-460, 474-476,

Response: We appreciate this constructive suggestion. The relevant content has now been relocated to the Discussion section as recommended.

None of the western blot results are quantified, with only letterbox example blots shown throughout. Please quantify all of these data. Without this, none of these results / conclusions can be accepted and it is very difficult to discuss further.

Line 421 from “immunoblotting”: This blot is very variable for control samples and without quantification cannot properly accept this..

Line 429: not quantified - this panel is not enough to convince of this.

Response: We appreciate the reviewer's suggestion. As recommended, we have performed quantitative analysis of all western blotting results and summarized in Supplementary Figure 8 of the revised manuscript (also attached below).

Line 490 -492 from “the ap2..” this should be in the methods section.

Response: We have moved this content to the Methods section as suggested.

Line 528: this crucial information should have been mentioned in the methods.

Response: As suggested by the reviewer, the Methods section has now been revised accordingly. Changes are highlighted in red in the revised manuscript.

The structure of the results section is confusing, going backwards and forwards between different mouse and cell models. Suggest to restructure, with all animal experiments coming before molecular / cell experiments, and showing all the AKO data first before over expression then cell data.

Response: We do understand the point raised by the reviewer regarding the structure of the Result section. The structure of our paper has now been re-organized as recommended: Figure 1-3 present resulted based on in vivo observations; Figure 4 presents expression of Netrin-1 in human samples/databases, and Figure 5-6 present data generated from mechanistic investigation.

The over-expression experiment is nice because the findings are in opposition to the KO model, thus supporting the role they initially describe.

Response: We would like to thank to reviewer for recognizing and acknowledging the value of our work.

Reviewer #2 (Remarks to the Author):

Figure 1 provides the only piece of evidence supporting adipocytes as a producer of Netrin-1, with poor quality of western blot, no quantification, and no validation with other methods on whether adipocytes are a key source of elevated serum Netrin-1 in obese and diabetic patients. After the generation of aKO mice, there were no data showing the contribution of adipocytes to its tissue and systemic levels.

Response: We appreciate the reviewer's concern. Regarding the quality of the immunoblot of Netrin-1, it is likely due to the fact that Netrin-1 is a secretory protein and given the difficulty of extracting protein from adipocytes, the images of western blotting experiments appeared to be perhaps less than ideal. Nevertheless, quantitative densitometry analysis of the immunoblot has now been provided and confirms significantly elevated Netrin-1 expression in adipocytes under high-fat diet conditions compared to normal diet controls, presented as Supplementary Figure 8 in the revised manuscript (also attached below).

Regarding whether Netrin-1 is a key source of Netrin-1, we cannot say for certain as the neuroendocrinological system is complex and worth extensive exploring. It is the main aim of the present study to investigate whether adipose-derived Netrin-1 would play a role on metabolism since we have inadvertently discovered difference in Netrin-1 expression between diabetic and healthy individuals (Figure 4D of the revised manuscript). Based on further investigation and results generated with the $NTN1^{akO}$ mice, it can be suggested that adipose-derived Netrin-1 at least plays a major role on maintaining metabolic parameters.

The authors argued that the improved insulin sensitivity may be explained by the increase in adipogenesis in inguinal but not other fat depots. There may be other possibilities such as altered adipose tissue secretome.

Response: We appreciate the reviewer's suggestion. A comprehensive metabolomics profiling of white adipose tissues obtained from the NTN^{AKO} mice has been performed as recommended and revealed minimal alterations in metabolite profiles following targeted ablation of Netrin-1. Figure is presented as Supplementary Figure 3E of the revised manuscript (also attached below).

The inhibitory effects of Netrin-1 on adipogenesis were suggested selective in inguinal fat. While the study using 3T3-L1 adipocytes as well as downstream targets PPAR γ and Wnt are well studied for white adipocytes in general.

Response: We thank the reviewer for this insightful comment. We do agree that the 3T3-L1 adipocytes have been used for in vitro mechanistic evaluation of an array of adipose functions, including that of inguinal fat^{1,2,3} (L.A. Pendergrast et al., Proc Natl Acad Sci USA. 2023, PMID: 36780527; B.Y. Li et al., Mol Metab. 2022, PMID: 35436587; V.J.M. Nies et al., Proc Natl Acad Sci USA. 2022). In the present study, we first isolated SVFs and confirmed an inhibitory effect of Netrin-1 on adipogenesis (Fig. 5D,H, I). To elucidate possible mechanisms that underline the effects of Netrin-1, we used the 3T3-L1 cells and identified the Wnt signaling pathway as a candidate (Fig. 5K). Besides, extracted inguinal SVFs from the NTN1^{AKO} mice were also used in parallel to confirm the role of Wnt signaling (Fig. 5L).

Regarding the seemingly “selective” expansion of the inguinal fat, this may be due to a lower Netrin-1 expression in the visceral precursor adipocytes (eWATs) compared to subcutaneous precursor adipocytes (iWATs) as reported in the database of Geotype-Tissue Expression Portal funded by the NIH of the USA (<https://gtexportal.org/home/gene/NTN1#geneExpression>) and also found in the present study (Supplementary Figure 3F). As a result, following genetic ablation of Netrin-1 in the adipose depots, expansion of the inguinal fat (having a significantly higher Netrin-1 expression level) may be more observable within the experimental period in our animal model.

In obesity, both visceral and inguinal fat expand with increased adipogenesis, which was not consistent with the summary in figure 7

Response: We appreciate your question and have revised the original Figure 7 accordingly and attached below.

Visceral fat even experiences higher levels of hypoxia than inguinal fat in obesity. Whether induced HIF-1 α leads to impaired adipogenesis of inguinal fat through Netrin-1-mediated interaction between adipocyte and preadipocyte under this condition requires further clarification.

Response: We appreciate and agree with the reviewer here. Indeed, published literature have indicated that visceral fats are more prone to hypoxia. Although, as mentioned earlier, due to the significant difference in Netrin-1 expression between the visceral and subcutaneous fat depots, genetic ablation of Netrin-1 may result in more observable changes in the subcutaneous fat depot as it expresses significantly higher levels of Netrin-1. Based on data presented in this study, we propose that under high-fat feeding condition, the induced high level of HIF-1 α *per se* could regulate the expression of Netrin-1 by binding to the Netrin-1 promoter region as an external stimuli in response to high-fat feeding (Fig.6A-F), which could then activate downstream signaling pathways such as the Wnt and PPAR γ (Fig. 5I-L).

We do agree that whether the HIF-1 α , with its myriad of downstream effects on adipocyte metabolism (reviewed in Huynh et al., J Mol Cell Biol. 2025) as well as on adipose immune cells (Li et al., Nat Immunol 2021; Engin A.B. Adv Exp Med Biol. 2024), could regulate adipogenesis via Netrin-1 indeed requires further investigation. Although given the already published literature, implementation of HIF-1 α ^{fl/fl} combined with the adipocyte-specific cre-promoter mice would ideally provide more systemic and comprehensive experimental evidence. It is therefore with regret that based on data generated in the present study (mainly investigating the role of adipose Netrin-1), no definitive suggestions regarding the role of HIF-1 α on adipogenesis (via Netrin-1 or not) can now be concluded. However, it is also our aim to investigate further as suggested by the reviewer.

Reviewer #3 (Remarks to the Author):

Major

comments:

Figure 2: data from only very few HFD mice (n=4) are presented. Are these data coming from one cohort of mice? Did authors confirm these finding in at least two other independent cohorts of mice showing the same significance? If yes, please add these data to the manuscript. If not, this needs to be done since in vivo data gained with an n of 4 mice per group may be underpowered.

Response: We understand the reviewer's concern. In fact, we have conducted multiple rounds of independent experiments before proceeding onto further mechanistic experiments. Data from the other 2 rounds of experiments (n=4-5) are now shown below.

Authors show only one H+E stained liver section to suggest that KO mice accumulate less lipids in livers. This is clearly not enough. Please quantify liver lipid content using a biochemical assay in livers of HFD-fed control and knockout mice. Similarly, please show pAkt from more than n=1 (Supplementary Figure 2) and quantify bands of Western blots to support the notion that adipocytes of KO mice show increased insulin signaling.

Response: Once again, we understand the reviewer's concern and appreciate the suggestions. We have now included quantitative data of hepatic lipid content as Supplementary Figure.2A (also attached below as A) as well as the adipocyte AKT signaling pathway western blotting data in Supplementary Fig. 2B (also attached below as B).

Figure 3: Authors show that Nestrin-1 overexpression decreases ORO in primary adipocytes (Fig. 3G/H) and PPAR γ expression in 3T3-L1 adipocytes. Are PPAR γ levels also decreased in primary adipocytes? Moreover, treatment of 3T3-L1 preadipocytes with Netrin-1 promotes β-catenin and authors conclude that "our results demonstrate that adipose Netrin-1 mitigate adipogenic differentiation of preadipocytes by activating the β-catenin pathway". However, such link is not shown. Authors need to repeat experiment in Nestin-1 overexpressing preadipocytes and/or in recombinant Netrin-1-treated preadipocytes after inhibiting/knowing down β-catenin and show that inhibition of the β-catenin pathway blunts the negative effect of Netrin-1 on adipogenesis.

**Response:** We appreciate the reviewer's comments. To clarify, we have overexpressed Netrin-1 in primary mouse preadipocytes (from SVFs) and performed Oil Red O staining, as now shown in Figure.5H. And observed decreased PPARγ protein levels following Netrin-1 overexpression and the induction of adipogenic differentiation (Fig. 5I, left panel). More importantly, as suggested by the reviewer, we have also performed experiments using primary mouse preadipocytes that overexpress Netrin-1 and exposed them to a β-catenin inhibitor, XAV939. The results (Supplementary Figure 7, also attached below) demonstrate that adipogenic differentiation toward mature adipocytes as indicated by red spots reduced in the Netrin-1(Ntn1)-treated group while the group exposed to Netrin-1 in the presence of β-catenin inhibitor XAV939 (Ntn1+XAV939) showed similar level of mature adipocytes to the control group.

Figure 5: Please analyze and quantify PPAR γ protein levels from several mice per group (Fig. 5E). Based on the hypothesis, one would expect that HFD-fed overexpressing mice reveal reduced fat mass/mass of iWAT (i.e. reduces WAT expansion) as well as increased liver lipid accumulation (lipid spillover, ectopic fat accumulation). Therefore, these parameters (WAT mass, liver lipid levels) need to be shown.

Response: As suggested by the reviewer, relative weights of the white adipose tissues and livers obtained from the AAV-administered mice as well as the hepatic lipid content were also analyzed, shown as Supplementary Figure 5.D-E (also attached below as A and C). We have also included the expression data of PPAR γ expression from more samples generated during the experimental observation (Figure 2H of the revised manuscript, also attached below as B). Indeed, our results revealed that under high-fat feeding condition, mice from the Netrin-1-overexpressing group showed no statistically significant changes of liver mass nor of the WATs, although increased liver lipid content was observed compared to the Control group.

Review3 Minor comments

Abstract, line 71: please adapt, since insulin resistance/sensitivity is not shown in overexpressing mice.

Response: We have now revised the manuscript according to the reviewer's comment. The changes are highlighted in red in the revised manuscript.

Supplementary Fig. 3E: quality of images is very poor and need to be improved. Please note that n=3 is shown in the graph but n=4 mice mentioned in the Figure legend. Please correct.

Response: We do apologize for our negligence and have now removed the images in

concern and corrected the incoherence regarding the number of samples used in the experiments.

Supplementary Fig. 6: please check mRNA expression of Netrin-1 in livers similar as done for WAT.

Response: We acknowledge the reviewer's concern and agree with the reviewer. Quantitative PCR was performed and the hepatic Netrin-1 expression is low, giving Ct values >34 across all samples.

Line 325-331; 346-353: please move to Discussion Please expand the discussion by adding the limitation of having only male (but not female mice) included in this study.

Response: According to the reviewer's suggestion, we have now revised the manuscript text with additional discussion on the limitation of this study. Changes are highlighted in Line 671-677 in the revised manuscript.

What is the background strain of *Ntn1^{fl/fl}* and *Ap2-Cre* mice? Why were *Ap2-Cre* rather than the more adipocyte-specific *AdipoqCre*-mice used in this study?

Response: To answer the reviewer's question, the mouse strain used for *Ntn1^{fl/fl}* and *Ap2-Cre* is C57BL/6J. It is with regret that at the time of this project, we were unable to obtain the *Adipoq-Cre* mice. Further investigation with the *AdipoqCre* or *PdgfraCre* would perhaps provide more information on the association between neuroendocrinological regulation and obesity.

Statistics: where data checked for normal distribution and similar variance? If not, this needs to be done and statistical tests need to be adapted in case of non-normal distribution/non-similar variance

Response: We have checked all data for normal distribution using statistical tests (Shapiro-Wilk/Kolmogorov-Sminov) in SPSS. The data sets included here were found to following a normal distribution before being subjected to further statistical analysis.

Response to the Reviewers' comments:

Reviewer #3 (Remarks to the Author)

In the abstract, authors still mention that "...WAT Netrin-1 overexpression used in vivo adeno-associated virus delivery resulted in worsened insulin resistance in both high fat and normal chow-fed mice". This is not appropriate since no data looking at insulin sensitivity (e.g., insulin tolerance test) are shown for overexpressing mice. Rather, a glucose tolerance test is presented, which is not only affected by insulin sensitivity but also insulin production. Please adapt the abstract accordingly.

Response: We thank the reviewer for pointing this out. We have now revised the description as "impaired glucose tolerance" in the revised manuscript (Abstract, line 70-73).

Statistics: authors did not reply do my question whether data that were compared had similar variance. If variance between groups was not similar, this needs to be taken into consideration when doing t-tests (Welch's correction).

Response: As suggested by the reviewer, all data were statistically analyzed using Graphpad Prism 10, with the in-built option of unpaired t-test with Welch's correction. Some detailed key statistical data are listed below as examples. Corresponding text in the regarding statistical methods were also highlighted in red in the revised manuscript (line 316-317) and in the Figure legends:

Fig.1E 45 min (HFD, GTT, ako vs WT): $P=0.0477$, Welch's correction $t=2.482$, $df=5.990$;

Fig.1E 60 min (HFD, GTT, ako vs WT): $P=0.0053$, Welch's correction $t=4.276$, $df=5.959$;

Fig.1E 120 min (HFD, GTT ako vs WT): $P=0.0409$, Welch's correction $t=3.284$, $df=3.267$.

Fig.1F 45 min (HFD, ITT ako vs WT): $P=0.0461$, Welch's correction $t=2.572$, $df=5.446$;

Fig.1F 60 min (HFD, ITT ako vs WT): $P=0.0053$, Welch's correction $t=4.315$, $df=5.850$;

Fig.1F 90 min (HFD, ITT ako vs WT): $P=0.0450$, Welch's correction $t=2.644$, $df=5.089$.

Fig.1J Body weight (HDF 4-8 weeks), $P=0.0185$, Welch's correction, $t=2.949$, $df=7.988$.

Fig.2C 15 min (GTT AAV-Ntn1 vs AAV-Control), $P=0.0215$, Welch's correction $t=2.864$, $df=7.837$;

Fig.2C 90 min (GTT AAV-Ntn1 vs AAV-Control), $P=0.0241$, Welch's correction $t=2.892$, $df=6.770$;

Fig.2C 120 min (GTT AAV-Ntn1 vs AAV-Control), $P=0.0171$, Welch's correction $t=3.248$, $df=6.101$.

Fig.2C 0 min (HFD, GTT AAV-Ntn1 vs AAV-Control), $P=0.0498$, Welch's correction $t=2.473$, $df=5.772$;

Fig.2C 45 min (HFD, GTT AAV-Ntn1 vs AAV-Control), $P=0.0103$, Welch's correction $t=5.356$, $df=3.254$;

Fig.2C 60 min (HFD, GTT AAV-Ntn1 vs AAV-Control), $P=0.0127$, Welch's correction $t=4.560$, $df=3.661$;

Fig.2C 90 min (HFD, GTT AAV-Ntn1 vs AAV-Control), $P=0.0074$, Welch's correction $t=4.342$, $df=5.001$.

Please clearly state the background strain of used Ntn1fl/fl and Ap2-Cre mice in the methods section.

Response: We appreciate the reviewer's suggestion and have now included the background information of the transgenic mouse strains in the Methods section (line 191-193). The background strain of both the Ntn1fl/fl and the aP2-cre mice was C57BL/6J as provided by the Shanghai Model Organisms Center Inc.